# 3D Aware Region Prompted Vision Language Model

**An-Chieh Cheng**[1] **Yang Fu**[1] **Yukang Chen**[3] **Zhijian Liu**[3] **Xiaolong Li**[3]
**Subhashree Radhakrishnan**[3] **Song Han**[2,3] **Yao Lu**[3] **Jan Kautz**[3] **Pavlo Molchanov**[3]
**Hongxu Yin**[3†] **Xiaolong Wang**[1†] **Sifei Liu**[3†]
[1]UC San Diego [2]MIT [3]NVIDIA

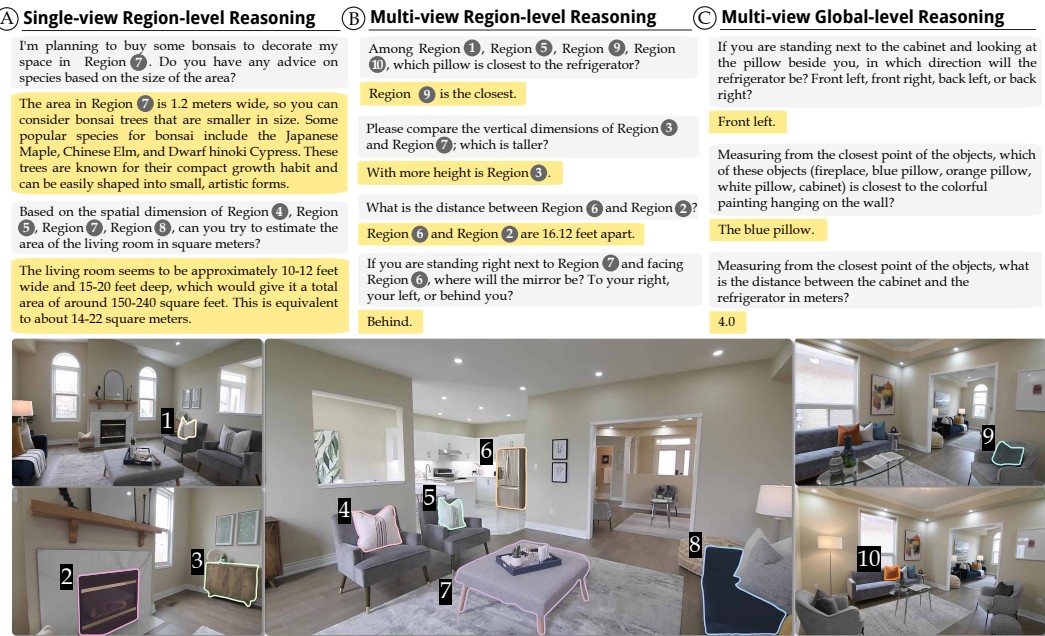

Figure 1: From precise region-based distance estimation (*left*), to intricate multi-view region query (*middle*), and global cross-frame reasoning (*right*), SR-3D delivers flexible and accurate spatial understanding to foundational Vision-Language Models. Notably, this video is obtained in the wild, **without sensory 3D inputs**, showcasing the remarkable generalization capability of our model.

https://www.anjiecheng.me/sr3d

## Abstract

We present **S**patial **R**egion **3D** (*SR-3D*) aware vision–language model that connects single-view 2D images and multi-view 3D data through a shared visual token space. SR-3D supports flexible region prompting, allowing users to annotate regions with bounding boxes, segmentation masks on any frame, or directly in 3D, without the need for exhaustive multi-frame labeling. We achieve this by enriching 2D visual features with 3D positional embeddings, which allows the 3D model to draw upon strong 2D priors for more accurate spatial reasoning across frames, even when objects of interest do not co-occur within the same view. Extensive experiments on both general 2D vision language and specialized 3D spatial benchmarks demonstrate that SR-3D achieves state-of-the-art performance, underscoring its effectiveness for unifying 2D and 3D representation space on scene understanding. Moreover, we observe applicability to in-the-wild videos without sensory 3D inputs or ground-truth 3D annotations, where SR-3D accurately infers spatial relationships and metric measurements.

## 1 Introduction

The rapid advancement of Vision Language Models (VLMs) (OpenAI, 2024; Liu et al., 2023; Anil et al., 2023; Wang et al., 2024b; Bai et al., 2025; Liu et al., 2025c) has demonstrated strong capabilities in visual understanding (Pratt et al., 2023; Huang et al., 2024a) and language grounding (Lv

---

† Equal advising.

et al., 2023). However, extending these strengths to 3D-aware spatial reasoning remains challenging. Foundational 2D VLMs excel at interpreting planar images, but generally lack mechanisms to capture complex 3D structural relationships. In contrast, most 3D VLMs (Hong et al., 2023; Huang et al., 2024c;b; Xu et al., 2024; Chen et al., 2024b) operate in a fundamentally different representation space, making it difficult to leverage the prior knowledge from foundational 2D VLMs. Their performance is often hindered by limited 3D training data. Moreover, specifying spatial relationships solely through language can be cumbersome in cluttered scenes, e.g., multiple objects of the same category can coexist. A more direct way of specifying object instances is highly desirable.

To mitigate these challenges, recent efforts adopt multi-view images as a 3D representation that aligns seamlessly with the input space of foundational 2D VLMs (Zhu et al., 2025; Zheng et al., 2025; Liu et al., 2025a). Unlike point clouds (Huang et al., 2024c;b; Xu et al., 2024) which require extensive data collection and model alignment, a multi-view approach leverages strong 2D priors for 3D scene understanding. To specify object instances during reasoning, region prompts have proven effective in single-view VLMs (Guo et al., 2024; Cheng et al., 2024; Yuan et al., 2024b; Rasheed et al., 2024). However, extending region prompting to multi-view settings remains challenging. Specifically, an object may appear across different views with varying visibility, making comprehensive multi-frame or 3D bounding box annotation tedious and text-based queries imprecise. Ideally, a practical VLM should allow straightforward region annotations, such as marking a bounding box on a single frame, while still accurately reasoning about spatial relationships across the entire multi-view scene.

Thus, we introduce SR-3D, a unified visual representation for 3D spatial understanding that leverages robust 2D foundational priors and supports flexible region prompting. In contrast to previous approaches that incorporate positional information only at 3D finetune stages (Zheng et al., 2025), or in different pathways (Zhu et al., 2025), we directly integrate positional embeddings within the foundational VLM. Specifically, we estimate each input image's depth using an off-the-shelf depth estimator (Yang et al., 2024) and transform this depth map into normalized 3D positional embeddings. For multi-view inputs representing a coherent scene, we further unify these positional embeddings into a common 3D coordinate space using either provided ground-truth camera poses or a point cloud estimator (Wang et al., 2024c; Leroy et al., 2024; Wang et al., 2025c) when only video inputs are available. Additionally, we incorporate region tokens directly into user prompts and train these region embeddings consistently at both the foundational single-view stages and the multi-view fine-tuning stage. Since the foundational VLM employs a dynamic tiling-based visual encoder (Chen et al., 2024d; Liu et al., 2025c), we design a novel branch specifically compatible with this architecture to produce robust region embeddings.

SR-3D naturally supports flexible region annotation on any frame. This stems from two design choices: (1) consistent 3D positional embeddings in a canonical space, enabling coherent cross-frame correspondences, and (2) an aligned embedding space from single-view pretraining that unleashes the full potential of region embeddings to generalize in multi-frame contexts. As evidence, our 2D-VLM trained only on single-view data demonstrates strong zero-shot spatial reasoning in 3D scenes, with and without region prompts, despite never seeing multi-view data.

We extensively evaluate across single-view and 3D multi-view settings, covering both region-level and global QA on general and spatial tasks. Our foundational 2D-VLM delivers large gains on region-level performance, surpassing prior state-of-the-art in both recognition and spatial understanding. These gains come without sacrificing general VQA accuracy and even improve tasks requiring spatial knowledge. With 3D fine-tuning, our model sets new state-of-the-art results in general 3D QA, video spatial reasoning, and region-level video tasks.

Our contributions are as follows:

- We introduce SR-3D, the first 3D-aware vision-language model that unifies representations for both single-view and multi-view tasks.
- We propose a dynamic tiling-based region extractor that handles high-resolution images and produces robust region embeddings. Our unified embedding space enables region representations trained on 2D images to generalize towards multi-view context.
- SR-3D achieves state-of-the-art results in general 3D QA, video spatial reasoning, and region-based video tasks, demonstrating strong generalization and scalability.

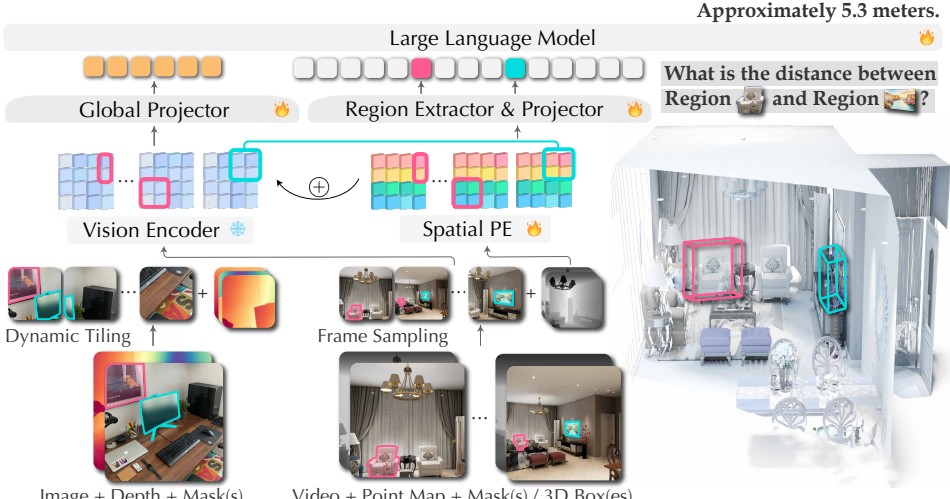

Figure 2: The SR-3D architecture. Given an image or multi-view input with optional region prompts (e.g., bounding boxes or masks), we encode them along with depth-derived positional embeddings using a tiling approach. Region tokens are extracted by stitching masked features, while 3D positional embeddings are mapped to a shared canonical space in the multi-view setting, as shown on the bottom right.

- We demonstrate real-world applications where our model effectively handles in-the-wild captured videos without 3D annotations (Figure 1), and can be flexibly prompted with region-level inputs.

## 2 METHODOLOGY

We propose a 3D-aware VLM architecture for single- and multi-view spatial understanding. Our approach adapts strong 2D priors by directly integrating 3D positional embeddings into visual representations, enabling accurate cross-frame reasoning. To further improve region-level grounding, we introduce a Dynamic Tiling-based Region Extractor, which works efficiently across both single- and multi-view inputs. As shown in Figure 2, the framework comprises a vision encoder, a 3D position encoding module, a region extractor, and an LLM backbone. In this section, we describe three main components: (1) canonical 3D positional representation (Sec. 2.1), (2) the region extractor (Sec. 2.2), and (3) the training paradigm (Sec. 2.3), along with inference pipeline (Sec. 2.4).

### 2.1 CANONICAL 3D POSITIONAL REPRESENTATION

The key idea of SR-3D is a canonical positional feature shared across single- and multi-view inputs. This unified representation unleashes the full potential of large-scale single-view pretraining, carrying its spatial priors into multi-view scenarios.

**Single-View Representation.** We begin by pretraining our foundational VLM on large-scale 2D images to establish strong visual-language priors. Given a single-view image $I$, we estimate its relative depth map $D$ using DepthAnythingV2 (Yang et al., 2024). We then compute a pixel-wise 3D position map in the camera coordinate system via back-projection, which is further canonicalized into a normalized world coordinate system. This canonicalization ensures that spatial information is expressed in a consistent and unified space, independent of camera pose.

To inject spatial information into VLM, we encode the corresponding 3D position map into embeddings using a sinusoidal function followed by a learnable point-wise MLP. These embeddings are resized to align with the token dimensions and then added to their respective vision tokens. This fusion enriches visual representations with geometric awareness, enabling the model to better capture object placement and spatial relationships within the scene.

**Multi-View Representation.** Building on the shared canonical space, we fine-tune the VLM with multi-view inputs to extend spatial reasoning beyond single images. We uniformly sample 32 frames from a video and resize the point maps to match the vision encoder's resolution. For multi-view training, we use ground-truth depth rather than estimated depth, performing back-projection and camera transformation to align the frames. The transformed point maps are normalized into the same canonical space as in the single-view setup, ensuring consistency in spatial representation. These

| Methods | Spatial | | | | Math |
| --- | --- | --- | --- | --- | --- |
| | BLINK$_S$ | SAT | EmbSpat | RealWorldQA | MathVista |
| NVILA-Lite-8B | 79.7 | 62.6 | 68.9 | 65.6 | 64.5 |
| **SR-3D-8B** | **83.9**$_{+4.2}$ | **64.0**$_{+1.4}$ | **72.5**$_{+3.6}$ | **68.1**$_{+2.5}$ | **65.4** |

| Methods | General Knowledge | | | | | OCR-Related | | |
| --- | --- | --- | --- | --- | --- | --- | --- | --- |
| | GQA | AI2D | MMMU$_p$ | SEED$_I$ | POPE | TextVQA | ChartQA | DocVQA |
| NVILA-Lite-8B | **65.3** | **91.0** | **25.1** | 76.3 | **88.1** | **78.1** | **84.8** | **91.7** |
| **SR-3D-8B** | 64.2 | 90.7 | 24.6 | **77.8** | 87.6 | 77.3 | 83.9 | 91.0 |

Table 1: Comparison of SR-3D and base model NVILA-Lite (Liu et al., 2025c)'s performance on general VQA benchmarks. SR-3D achieves stronger results on spatial-related benchmarks without compromising performance on general and OCR benchmarks, suggesting that 3D-aware pre-training enhances spatial reasoning while preserving the base model's broader knowledge.

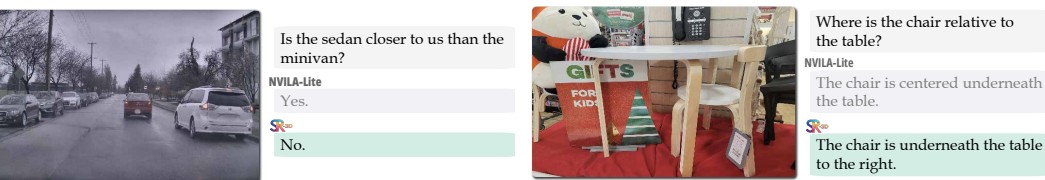

Figure 3: RealWorldQA (xAI, 2024) results. SR-3D shows stronger spatial understanding of physical environments compared to the base model. We omit the answer choices for clarity in visualization.

processed frames and point maps act as the multi-view analog of the single-view tiles, enabling seamless integration of spatial and visual information across both training stages.

## 2.2 DYNAMIC TILING-BASED REGION EXTRACTOR

**Tiling-based Encoder.** The visual backbone produces a low-resolution feature map, limiting its ability to represent small-scale regions and objects. To address this, we adopt the dynamic tiling mechanism employed in (Liu et al., 2025c) that enables high-resolution processing while maintaining spatial consistency. Instead of resizing entire images, we first determine the optimal aspect ratio by selecting the closest match from a predefined set (e.g., 1:1, 1:2, 2:1, 3:1, ..., 2:6), minimizing distortions. We then resize both the image and any corresponding point map accordingly and divide them into tiles of $448 \times 448$, matching the vision encoder's resolution. Each tile is encoded separately before being stitched back together, preserving local details without exceeding memory constraints. This tiling process is applied similarly to point maps and region masks, forming the basis for both our 3D positional embedding and region feature extraction strategies.

**Dynamic Region Extractor.** Prior architectures without dynamic tiling use feature refinement with deconvolution layers to upsample visual tokens (Cheng et al., 2024; Guo et al., 2024), aiming to recover lost details. But since this occurs after the vision encoder, where features are already resized and potentially distorted, the recovery of fine details is limited.

To address this, we introduce a *tile-then-stitch* approach to extract region embeddings from high-resolution features. For single-view input, given a region of interest (RoI) represented by a binary mask, we apply the same dynamic tiling process used in the image pipeline to generate tiles of both the image and the mask. The tiled visual tokens and masks are then stitched back together at a higher resolution, followed by a mask-pooling operation to obtain the final mask feature. This method offers two key advantages: (1) the extracted mask feature is derived from high-resolution features directly, reducing distortion and eliminating the need for post-refinement, and (2) our tile-then-stitch approach extends naturally to multi-view video inputs. In the multi-view setting, each frame is treated as a tile, allowing us to handle one or multiple masks per frame while maintaining spatial consistency across frames for the same RoI.

## 2.3 TRAINING PARADIGM

For the single-view VLM, we initialize the weights from a pre-trained 2D VLM (NVILA-Lite-8B (Liu et al., 2025c)), keeping the vision encoder frozen while fine-tuning the 3D positional encoding module, projectors, and the LLM. We reuse the instruction fine-tuning dataset from the pre-trained VLM and blend it with region-prompted datasets (Guo et al., 2024; Cheng et al., 2024) in this stage, resulting in a total data blend of approximately 7 million samples. Full dataset details are provided in the Supplementary Materials.

| Methods | Acc. (%) |
|---|---|
| Human | 98.3 |
| ***Proprietary Models (API)*** | |
| Gemini Pro (Anil et al., 2023) | 50.0 |
| Claude 3 OPUS (Anthropic, 2024) | 57.3 |
| GPT-4V-*Turbo* (OpenAI, 2024) | 66.9 |
| GPT-4o (OpenAI, 2024) | 64.5 |
| ***Open-source Models*** | |
| Yi-VL-34B (Young et al., 2024) | 53.2 |
| LLaVA-v1.5-13B-xtuner (Contributors, 2023) | 54.0 |
| LLaVA-v1.6-34B (Zhang et al., 2024b) | 64.5 |
| InstructBLIP-7B (Dai et al., 2023) | 50.8 |
| LLaVA-v1.5-7B-xtuner (Contributors, 2023) | 50.8 |
| LLaVA-v1.5-7B (Liu et al., 2024) | 51.6 |
| LLaVA-InternLM2-7B (Cai et al., 2024) | 52.4 |
| SpatialRGPT-8B (Cheng et al., 2024) | 87.9 |
| **SR-3D-8B** | **90.3** |

Table 2: Results on BLINK$_{\text{Depth}}$. We follow Cheng et al. (2024)'s protocol to test whether a VLM effectively leverages auxiliary 3D information.

| Methods | mAP ($\uparrow$) | Acc. (%) |
|---|---|---|
| CLIP (Radford et al., 2021) | 58.9 | - |
| RegionCLIP (Zhong et al., 2022) | 58.3 | - |
| LLaVA-7B (Liu et al., 2023) | - | 40.0 |
| Shikra-7B (Chen et al., 2023c) | - | 53.9 |
| GPT4RoI-7B (Zhang et al., 2023) | - | 64.0 |
| PVIT-7B (Chen et al., 2023a) | - | 64.5 |
| ASM-7B (Wang et al., 2024g) | 69.3 | - |
| RegionGPT-7B (Guo et al., 2024) | 70.0 | 80.6 |
| DynRefer (Zhao et al., 2025) | - | 81.2 |
| SpatialRGPT-8B (Cheng et al., 2024) | 72.9 | 82.9 |
| **SR-3D-8B** | **78.0** | **88.6** |

Table 3: Region-level classification results on COCO-2017 val set with ground-truth boxes, following RegionCLIP (Zhong et al., 2022) and RegionGPT (Guo et al., 2024).

For the multi-view model, we fine-tune the single-view model using datasets such as ScanQA (Azuma et al., 2022), SQA3D (Ma et al., 2023), and Scan2Cap (Chen et al., 2021), as well as a newly curated EmbodiedScan (Wang et al., 2024d) dataset with region- and spatial-focused question-answer pairs. To enhance robustness and generalization, we apply various mask augmentations during multi-view training, including converting segmentation masks into bounding boxes and randomly dropping frames to simulate single-frame annotations. These strategies help the model learn to associate regions across frames while preserving spatial consistency.

We note that, unlike prior work (Zhu et al., 2025) that employs separate pathways for single- and multi-view data, we adopt a unified pipeline where all data flows through the same model architecture. This ensures consistent processing of both single- and multi-view inputs without distinction between spatial region prompts and global queries.

## 2.4 INFERENCE

Our tile-and-stitch design enables flexible region-based inference. For single-view inputs, the model accepts bounding boxes or segmentation masks as region annotations. In multi-view scenarios, it supports a range of mask specifications: 3D bounding boxes that project into multi-frame masks, sparse-frame masks, or even a single-frame mask. This reflects SR-3D 's ability to handle varying annotation densities while preserving spatial alignment. For 3D multi-view input, although ground-truth depth maps were used during multi-view training, our approach remains highly adaptable due to the canonicalization of 3D positions into a normalized space. This allows us to replace ground-truth depth with point maps estimated from off-the-shelf models such as MAST3R (Leroy et al., 2024) or CUT3R (Wang et al., 2025c). Our model offers a highly flexible and generalizable solution for spatial reasoning across diverse input modalities by maintaining a unified architecture that normalizes spatial information across different 3D sources.

## 3 EXPERIMENTS

We first evaluate SR-3D on 2D benchmarks (Section 3.1) to verify whether the introduced positional features improve performance while preserving the generalization of the base single-view model. We then evaluate the multi-view model on 3D benchmarks in Section 3.2. Finally, we show ablation studies in Section 3.4 to analyze the role of pretraining and 3D positional encoding.

## 3.1 EVALUATION ON 2D BENCHMARKS

**Region-level Question Answering.** We evaluate our model's object classification performance on the COCO-2017 (Lin et al., 2014) dataset using mean Average Precision (mAP) and classification accuracy as metrics. Following prior work on region-level recognition (Zhong et al., 2022; Guo et al., 2024; Cheng et al., 2024), we rely on ground-truth boxes for positional information and augment the general prompt with task-specific instructions. As reported in Table 3, SR-3D attains an mAP of 78.0 and an accuracy of 88.6%, demonstrating strong region-level recognition and validating the effectiveness of our region extractor. Compared with SpatialRGPT (Cheng et al., 2024), which is trained on the same region-level data, our model achieves significant gains, largely attributable to the dynamic tiling extractor that provides higher-fidelity regional masks. For reference, we also include

| Methods | Scan2Cap | | | | ScanQA | | | | | SQA3D |
| --- | --- | --- | --- | --- | --- | --- | --- | --- | --- | --- |
| | B-4 ↑ | Rouge ↑ | Cider ↑ | Meteor ↑ | B-4 ↑ | Rouge ↑ | Cider ↑ | Meteor ↑ | EM ↑ | EM ↑ |
| *Task-specific Specialist* | | | | | | | | | | |
| VoteNet+MCAN (Yu et al., 2019) | - | - | - | - | 6.2 | 29.8 | 54.7 | 11.4 | 17.3 | - |
| ScanRefer+MCAN (Yu et al., 2019) | - | - | - | - | 7.9 | 30.0 | 55.4 | 11.5 | 18.6 | - |
| ScanQA (Azuma et al., 2022) | - | - | - | - | 10.1 | 33.3 | 64.9 | 13.1 | 21.0 | - |
| 3D-VisTA (Zhu et al., 2023) | 34.0 | 54.3 | 66.9 | 27.1 | 10.4 | 35.7 | 69.6 | 13.9 | 22.4 | - |
| *2D Large Multi-modal Models* | | | | | | | | | | |
| Oryx-34B (Liu et al., 2025d) | - | - | - | - | - | 37.3 | 72.3 | 15.0 | - | - |
| NaviLLM (Zheng et al., 2024) | - | - | - | - | 12.0 | 38.4 | 75.9 | 15.4 | 23.0 | - |
| LLaVA-Video-7B[†] (Zhang et al., 2025) | - | - | - | - | 3.1 | 44.6 | 88.7 | 17.7 | - | - |
| NaVILA (Cheng et al., 2025) | - | - | - | - | 16.9 | 49.3 | 102.7 | 20.1 | 28.6 | - |
| *3D Large Multi-modal Models* | | | | | | | | | | |
| 3D-LLM$_{(flamingo)}$ (Hong et al., 2023) | - | - | - | - | 7.2 | 32.3 | 59.2 | 12.2 | 20.4 | - |
| 3D-LLM$_{(BLIP2-flant5)}$ (Hong et al., 2023) | - | - | - | - | 12.0 | 35.7 | 69.4 | 14.5 | 20.5 | - |
| LL3DA (Chen et al., 2024b) | 36.8 | 55.1 | 65.2 | 26.0 | 13.5 | 37.3 | 76.8 | 15.9 | - | - |
| Chat-3Dv2 (Wang et al., 2023b) | - | - | - | - | 14.0 | - | 87.6 | - | - | 54.7 |
| LEO (Huang et al., 2024c) | 36.9 | 57.8 | 68.4 | 27.7 | 13.2 | 49.2 | 101.4 | 20.0 | 24.5 | 50.0 |
| Scene-LLM (Fu et al., 2024a) | - | - | - | - | 12.0 | 40.0 | 80.0 | 16.6 | 27.2 | 54.2 |
| ChatScene (Huang et al., 2024b) | 36.3 | 58.1 | 77.2 | 28.0 | 14.3 | 41.6 | 87.7 | 18.0 | 21.6 | 54.6 |
| LLaVA-3D (Zhu et al., 2025) | 41.1 | 63.4 | 79.2 | 30.2 | 14.5 | 50.1 | 91.7 | 20.7 | 27.0 | 55.6 |
| Video-3D LLM (Zheng et al., 2025) | 42.4 | 62.3 | 83.8 | 28.9 | 16.2 | 49.0 | 102.1 | 19.8 | 30.1 | 58.6 |
| **SR-3D-8B** | **44.7** | **67.3** | **97.9** | **31.5** | **18.1** | **51.2** | **109.3** | **21.2** | **30.4** | **62.2** |

Table 4: Evaluation of spatial scene understanding on the Scan2Cap, ScanQA, and SQA3D benchmarks.
[†] indicates methods evaluated in a zero-shot setting. SR-3D achieves state-of-the-art results across all metrics.

DynRefer's RoIAlign (448 variant) (Zhao et al., 2025) as a baseline at the same resolution. Their proposed strategies are also complementary to our approach.

We further evaluate SR-3D on the BLINK$_{Depth}$ benchmark (Fu et al., 2024b) using the region-prompts as in SpatialRGPT (Cheng et al., 2024), which tests point-level depth understanding in VLMs. BLINK$_{Depth}$ is a challenging task that requires both spatial and regional awareness. We report results in Table 2 showing that SR-3D outperforms current state-of-the-art SpatialRGPT (Cheng et al., 2024), achieving 90% accuracy. These results highlight that our approach excels in region extraction and effectively utilizes the provided 3D-aware input.

**General Question Answering.** We investigate two key questions: (1) Does incorporating 3D positional information affect general vision-language understanding capabilities? (2) Can it improve performance on spatial-related tasks? To answer these, we evaluate our model on general VLM benchmarks covering Spatial (xAI, 2024; Ray et al., 2025; Du et al., 2024; Fu et al., 2024b), Math (Lu et al., 2024), General Understanding (Hudson & Manning, 2019; Kembhavi et al., 2016; Yue et al., 2024; Li et al., 2024; 2023), and OCR-related (Singh et al., 2019; Masry et al., 2022; Mathew et al., 2021) tasks. As shown in Table 1, compared to the base model NVILA-Lite-8B (Liu et al., 2025c), our model maintains comparable performance in math, general understanding, and OCR-related tasks, confirming that integrating 3D positional information does not degrade overall vision-language capabilities. Additionally, our method improves performance on the spatial understanding benchmark RealWorldQA (xAI, 2024). We also provide qualitative examples from RealWorldQA in Figure 3, showcasing cases where NVILA-Lite fails while SR-3D succeeds. These results demonstrate that SR-3D enhances spatial reasoning while preserving general capabilities.

## 3.2 EVALUATION ON 3D BENCHMARKS

**General 3D Question Answering.** We report results on three classic 3D vision-language understanding tasks: 3D dense captioning on Scan2Cap (Chen et al., 2021), ScanQA (Azuma et al., 2022), and SQA3D (Ma et al., 2023) in Table 4. Our evaluation metrics include conventional scores (e.g., CIDEr, BLEU, METEOR, ROUGE) as well as exact-match (EM) accuracy. Following prior work, we assume that input scenes may lack 3D object mask annotations during inference and use off-the-shelf models to generate proposals. However, unlike previous approaches, we leverage 2D segmentation models to generate 2D object proposals instead. We compare SR-3D against strong baselines, including task-specific specialist models for each benchmark and leading methods from both 2D and 3D large multimodal models (LMMs). SR-3D significantly outperforms state-of-the-art single-task and task-specific fine-tuned models on 3D dense captioning and 3D QA tasks. Our design operates in a canonicalized 3D space, making it naturally compatible with geometric foundation models and well-suited for extending to casual in-the-wild videos. Motivated by this, we further evaluate robustness under reconstructed inputs using Cut3R (Wang et al., 2025c), comparing results against ground-truth point clouds. As shown in Table 9, SR-3D maintains performance close

| Methods | Wide/Thin | Tall/Short | Big/Small | Multi. Simple | Multi. Complex | Avg. | Width | Height | Distance | Avg. |
|---|---|---|---|---|---|---|---|---|---|---|
| | | | | Qualitative | | | | | Quantitative | |
| ***Blind LLMs w/ Language Referral*** | | | | | | | | | | |
| GPT-4o (OpenAI, 2024) | 64.8 | 64.5 | 64.0 | 47.8 | 41.4 | 56.5 | 70.5 | 70.6 | 50.4 | 63.8 |
| ***VLMs w/ Language Referral*** | | | | | | | | | | |
| GPT-4o (OpenAI, 2024) | 52.1 | 54.1 | 57.5 | 62.4 | 42.4 | 53.7 | 72.4 | 72.8 | 55.8 | 67.0 |
| NVILA-Video-8B (Liu et al., 2025c) | 48.8 | 38.9 | 53.7 | 52.1 | 36.0 | 45.9 | 59.2 | 54.3 | 6.6 | 40.0 |
| ***Region VLMs*** | | | | | | | | | | |
| GPT-4o (OpenAI, 2024)+SoM | 46.1 | 39.9 | 39.3 | 52.1 | 43.2 | 44.1 | 52.4 | 47.8 | 40.0 | 46.7 |
| NVILA-Video-8B (Liu et al., 2025c)+SoM | 49.3 | 40.0 | 53.7 | 52.1 | 40.4 | 47.1 | 59.3 | 54.1 | 6.6 | 40.0 |
| **SR-3D-8B** | **76.3** | **83.1** | **81.8** | **80.3** | **76.0** | **79.5** | **87.7** | **87.3** | **74.8** | **83.3** |

Table 5: Evaluation of region-level spatial scene understanding on the SR-3D-Bench. SR-3D outperforms all baselines, highlighting the importance of strong region understanding and spatial awareness. Notably, SoM struggles with multi-frame inputs, reflecting the inherent difficulty of multi-frame visual grounding.

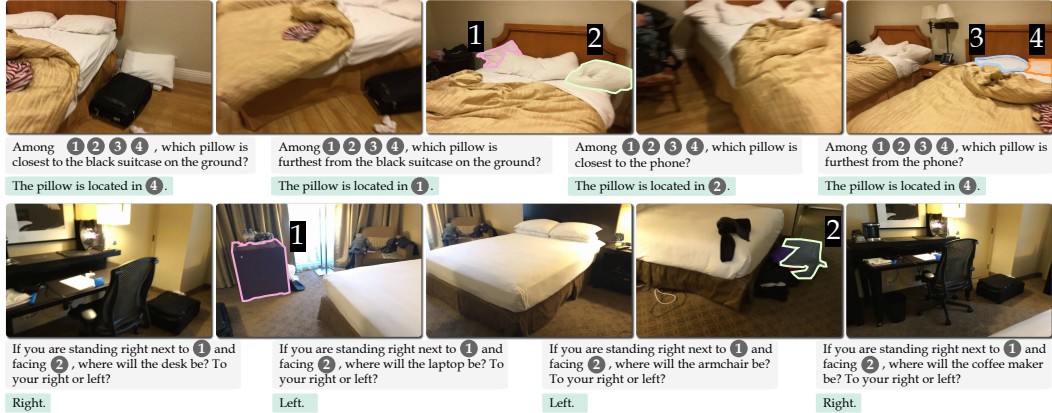

Figure 4: SR-3D results on region-level multi-view spatial understanding. We show extreme cases where the same region prompts are used across samples but with different target objects. SR-3D answers all queries correctly, showing strong evidence that it truly understands 3D spatial relationships. Note that the overlayed region ID tags are only for visualization in the paper to improve readability and are not used during inference.

to ground truth, highlighting its resilience to reconstruction artifacts, while Video3dLLM (Zheng et al., 2025) exhibits a clear performance drop.

## 3.3 VIDEO SPATIAL INTELLIGENCE.

**Region-level Spatial QA.** Currently, no video benchmarks specifically focus on region-level spatial understanding. Without explicit region information, spatial understanding can become ambiguous, especially when multiple identical objects are present or when referring to a specific area in a scene that is difficult to describe precisely using language alone. To address this, we propose SR-3D-Bench, a region-level spatial benchmark curated from ScanNet (Dai et al., 2017), ARKitScenes (Baruch et al., 2021), and Matterport (Chang et al., 2017) video scan datasets with 3D ground truth. Specifically, we utilize preprocessed oriented bounding box annotations from EmbodiedScan (Wang et al., 2024d), where each object is axis-aligned within a canonicalized geodetic coordinate system. This alignment ensures that the bounding box dimensions accurately represent the true width, length, and height. Using these bounding boxes, we construct a conversational benchmark that includes both qualitative and quantitative question-answering tasks. The qualitative QA consists of choice-based, predicate-based, and multiple-choice questions, while the quantitative QA focuses on measuring object width, height, and distance. We generate these QA pairs using template-based conversation generation and allow the VLM to generate free-form language. For qualitative QA evaluation, we use GPT-4o (OpenAI, 2024) as an evaluator and report the accuracy, while for quantitative QA, we measure the success rate by thresholding the maximum ratio between estimation and the ground truth value.

We report three types of baseline models: (1) Blind LLMs, which answer questions using only the provided text without visual input. To improve this, we replace the mask prompt with the object class

| Methods | Obj. Count | Abs. Dist. | Obj. Size | Room Size | Rel. Dist. | Rel. Dir. | Route Plan | Appr. Order | Avg. |
|---|---|---|---|---|---|---|---|---|---|
| | Quantitative | | | | Qualitative | | | | Avg. |
| Random | - | - | - | - | 25.0 | 36.1 | 28.3 | 25.0 | - |
| Human Level† | 94.3 | 47.0 | 60.4 | 45.9 | 94.7 | 95.8 | 95.8 | 100 | 79.2 |
| *Proprietary Models (API)* | | | | | | | | | |
| GPT-4o (OpenAI, 2024) | 46.2 | 5.3 | 43.8 | 38.2 | 37.0 | 41.3 | 31.5 | 28.5 | 34.0 |
| Gemini-1.5 Flash (Georgiev et al., 2024) | 49.8 | 30.8 | 53.5 | 54.4 | 37.7 | 41.0 | 31.5 | 37.8 | 42.1 |
| Gemini-1.5 Pro (Georgiev et al., 2024) | **56.2** | 30.9 | 64.1 | 43.6 | 51.3 | 46.3 | **36.0** | 34.6 | 45.3 |
| *Open-source Models* | | | | | | | | | |
| InternVL2-8B (Chen et al., 2024e) | 31.3 | 29.0 | 48.9 | 44.2 | 38.0 | 33.4 | 28.9 | 46.4 | 37.5 |
| InternVL2-40B (Chen et al., 2024e) | 41.3 | 26.2 | 48.2 | 27.5 | 47.6 | 32.7 | 27.8 | 44.7 | 37.0 |
| LongVILA-8B (Xue et al., 2025) | 29.1 | 9.1 | 16.7 | 0.0 | 29.6 | 30.7 | 32.5 | 25.5 | 21.6 |
| VILA-1.5-8B Lin et al. (2024) | 17.4 | 21.8 | 50.3 | 18.8 | 32.1 | 34.8 | 31.0 | 24.8 | 28.9 |
| VILA-1.5-40B Lin et al. (2024) | 22.4 | 24.8 | 48.7 | 22.7 | 40.5 | 25.7 | 31.5 | 32.9 | 31.2 |
| LongVA-7B (Zhang et al., 2024a) | 38.0 | 16.6 | 38.9 | 22.2 | 33.1 | 43.3 | 25.4 | 15.7 | 29.2 |
| LLaVA-Video-7B (Zhang et al., 2024b) | 48.5 | 14.0 | 47.8 | 24.2 | 43.5 | 42.4 | 34.0 | 30.6 | 35.6 |
| LLaVA-Video-72B (Zhang et al., 2024b) | 48.9 | 22.8 | 57.4 | 35.3 | 42.4 | 36.7 | 35.0 | 48.6 | 40.9 |
| LLaVA-OneVision-7B (Li et al., 2025) | 47.7 | 20.2 | 47.4 | 12.3 | 42.5 | 35.2 | 29.4 | 24.4 | 32.4 |
| LLaVA-OneVision-72B (Li et al., 2025) | 43.5 | 23.9 | 57.6 | 37.5 | 42.5 | 39.9 | 32.5 | 44.6 | 40.2 |
| **SR-3D-8B** | 54.9 | **53.8** | **74.5** | **65.1** | **63.5** | **81.8** | 33.5 | **75.9** | **62.9** |

Table 6: Results on multi-view global spatial scene understanding evaluated on VSI-Bench (Yang et al., 2025b). † indicates methods tested on the `Tiny` subset. SR-3D achieves strong performance, especially on the relative direction task, providing clear evidence that the model effectively leverages the 3D positional encoding.

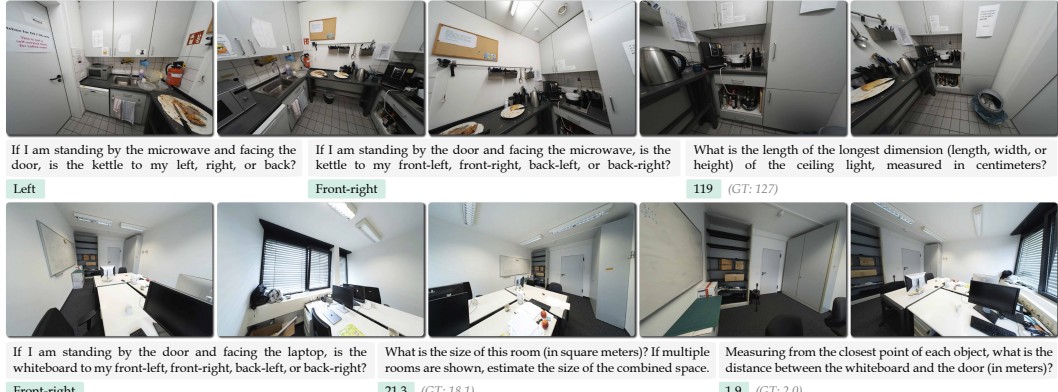

Figure 5: VSI-Bench (Yang et al., 2025b) results. SR-3D answers spatial questions correctly even without region prompts, handles fine-grained directional queries such as distinguishing `front-left` from `front-right`, and accurately answers metric-scale spatial questions like distance queries.

for each question. This serves as a baseline to measure how much video spatial reasoning can come from general world knowledge alone. We use GPT-4o (OpenAI, 2024) as the representative, as it is one of the most advanced models for general knowledge. (2) VLMs with Language Referral, which have access to visual content, allowing them to potentially perform better than blind LLMs. We use state-of-the-art vision-language models GPT-4o and NVILA-Video (Liu et al., 2025c) as baselines in this category. (3) Region-aware Video VLMs. These models process specific image regions without relying on text descriptions or object class information. We equip GPT-4o and NVILA-Video with Set of Marks (SoM) for region-based reasoning. Note that while Qiu et al. (2024) and Wang et al. (2024a) are also region-level video VLMs, they are excluded from comparisons as they cannot handle multi-object input or lack support for multi-frame prompts.

We present results in Table 5. The findings suggest that both Blind LLMs and VLMs with Language Referral perform reasonably well on quantitative tasks, such as estimating object width, due to their general world knowledge. However, region-level VLMs equipped with SoM struggle, likely because the models find it challenging to track the set of marks across frames. Overall, our method outperforms all baselines across all categories.

**Global Spatial QA.** We also report results on global spatial understanding using VSI-Bench (Yang et al., 2025b), a recently proposed benchmark that quantitatively evaluates the visual-spatial intelligence of VLMs based on egocentric videos. To avoid potential effects from noisy or inconsistent labels, training samples from the ScanQA series are excluded. We follow the original setting and

| | 2D Pre-train | 3D Tall/Short | 3D Big/Small | 3D Height | 3D Distance |
|---|---|---|---|---|---|
| *Zero-shot 2D Models* | | | | | |
| Base Model | | $40.0_{-31.4}$ | $53.7_{-26.0}$ | $54.1_{-14.4}$ | $6.6_{-61.9}$ |
| **SR-3D-2D** | ✓ | **71.4** | **79.7** | **68.5** | **68.5** |
| *Finetuned 3D Models* | | | | | |
| SR-3D | | $83.1_{-0.0}$ | $80.5_{-1.3}$ | $85.7_{-1.6}$ | $60.3_{-14.5}$ |
| **SR-3D** | ✓ | **83.1** | **81.8** | **87.3** | **74.8** |

Table 7: Zero-shot evaluation of our 2D-trained VLM on SR-3D-Bench, testing whether the model's representations are truly aligned. SR-3D-2D achieves reasonable accuracy without explicit 3D supervision.

| 3D PE | PT | Scan2Cap | ScanQA | SQA3D | 3D Region | 3D Global |
|---|---|---|---|---|---|---|
| | | 92.9 | 101.3 | 58.6 | 74.0 | 51.1 |
| | ✔ | 94.3 | 108.2 | 59.5 | 78.1 | 52.9 |
| ✔ | | 92.7 | 102.9 | 59.1 | 75.3 | 51.2 |
| ✔ | ✔ | **97.9** | **109.3** | **62.2** | **80.9** | **62.0** |

Table 8: Ablation study on the impact of incorporating 3D positional embeddings (3D PE) and single-view pre-training (PT). The results indicate that both 3D positional embeddings and single-view pre-training are crucial, and further scaling up pre-training is likely to yield additional gains.

| | 3D Source | C ↑ | B-1 ↑ | B-4 ↑ | M ↑ | R ↑ | EM ↑ |
|---|---|---|---|---|---|---|---|
| Video-3D LLM | GT | 102.1 | 47.1 | 16.2 | 19.8 | 49.0 | 30.1 |
| Video-3D LLM | Cut3R | 100.7 | 46.6 | 15.8 | 19.6 | 48.6 | 29.9 |
| SR-3D | GT | 109.3 | 50.9 | 18.1 | 21.2 | 51.2 | 30.4 |
| **SR-3D** | **Cut3R** | **109.3** | **50.9** | **18.1** | **21.2** | **51.2** | **30.2** |

Table 9: ScanQA results on ground-truth and Cut3R-reconstructed point clouds, compared with Video-3D LLM (Zheng et al., 2025). SR-3D exhibits a smaller performance drop than the baseline when shifting from ground-truth to reconstructed inputs.

use accuracy as the evaluation metric for qualitative questions and Mean Relative Accuracy (MRA) for quantitative questions. As shown in Table 6, SR-3D outperforms all open-source models and performs comparably, if not better, than API-based models.

### 3.4 ANALYSIS AND ABLATION STUDY

**Zero-shot Generalization.** In this analysis, we ask: Can a foundational 2D VLM trained only on single-view images perform zero-shot spatial reasoning on multi-view 3D scenes? To test this, we evaluate its zero shot performance on SR-3D-Bench across the Tall/Short, Big/Small, Height, and Distance categories. We exclude width because it is defined differently in single-view and multi-view settings: in single-view images, it refers to the horizontal extent in the image plane (Cheng et al., 2024), while in multi-view scenes, it denotes the maximum object dimension. Table 7 presents the results, showing that the single-view model performs strongly. This indicates that our unified representation transfers knowledge from single-view images effectively, even though the model has not seen multi-view data, scene-level position embeddings, or ground truth spatial annotations.

**3D Position Embedding and Single-view Pre-training.** We conduct an ablation study to evaluate the impact of single-view pre-training and 3D positional embeddings. Four model variants are compared, with/without pre-training and with/without 3D positional embeddings. As shown in Table 8, single-view pre-training provides substantial gains by allowing the model to transfer spatial knowledge, while 3D embeddings offer limited improvements at the current scale. These findings highlight the need for larger-scale settings to fully exploit positional representations for spatial reasoning.

## 4 RELATED WORK

Our work builds upon recent advancements in region-level understanding (Yuan et al., 2024b; Guo et al., 2024), spatial reasoning (Chen et al., 2024a; Yang et al., 2025b), and 3D large multimodal models (Hong et al., 2023; Chen et al., 2024b). The most closely related methods are LLaVA-3D (Zhu et al., 2025) and Video-3D LLM (Zheng et al., 2025), which also integrate 3D position-aware features into 2D VLMs. However, these approaches often rely on separate processing pathways for 2D/3D data or require fine-tuning on specialized 3D video data, which risks overfitting position encodings to specific tasks. In contrast, we propose a unified architecture and a shared 3D representation space for both images and videos, fostering better alignment and improving generalization across spatial understanding tasks. A comprehensive literature review is in Appendix B.

## 5 CONCLUSION

We introduce SR-3D, a foundational vision language model for 3D-aware spatial reasoning. By unifying single- and multi-view data, our approach adapts strong 2D priors from pretrained VLMs into a 3D-aware representation for complex spatial tasks. Additionally, our tile-and-stitch method extracts high-resolution region features, enabling flexible region prompts. Experiments on 2D and 3D benchmarks show state-of-the-art performance, validating SR-3D's ability to unify and enhance spatial reasoning, unlocking the potential of 3D-aware VLMs.

## ACKNOWLEDGEMENT

This project was supported, in part, by NSF CAREER Award IIS-2240014, gifts from Amazon, Meta, and Qualcomm.

## ETHICS STATEMENT

SR-3D is developed as a general-purpose visual assistant, similar to other vision language models (OpenAI, 2024; xAI, 2024; Georgiev et al., 2024). While it offers potential benefits for tasks in robotics, AR/VR, and other domains, it also shares common concerns associated with large language and multimodal models. These include the risk of output hallucinations, inherited biases from pretrained models, and the environmental impact of scaling to larger architectures. Evaluating spatial reasoning performance remains challenging (Cheng et al., 2024), and further research is needed to ensure robustness and reliability, particularly in safety-critical domains such as robotics. In potential applications to VR/AR smart glasses, future work should also address privacy and security concerns. Our work serves as a research prototype, and we do not claim deployment readiness. No human subjects were involved in this study, and no personally identifiable information was collected. The supplementary demonstration video on the website uses publicly available YouTube footage and is provided solely for academic research purposes, not for commercial use.

## REPRODUCIBILITY STATEMENT

We have taken several measures to ensure the reproducibility of our work. SR-3D builds upon an open-sourced vision language model (Liu et al., 2025c) as the base, and all datasets used in our experiments are publicly available, with no in-house or proprietary data involved. In the main paper and appendix, we provide detailed descriptions of the data curation pipeline, model architecture, and training hyperparameters. To further support reproducibility, we will release our curated datasets, benchmark, source code, and pretrained model weights as open-sourced software.

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

APPENDIX: TABLE OF CONTENTS

# A APPLICATIONS

Our method is flexible in two key ways. First, since SR-3D is trained in a normalized 3D space, it naturally integrates with existing 3D foundation models (Wang et al., 2025c; Leroy et al., 2024; Wang et al., 2025d;b) for pointmap estimation. This design allows the input to extend beyond 3D scans, and SR-3D can also process in the wild videos such as YouTube footage. Second, SR-3D removes the need for costly 3D annotations or dense per-frame labeling. Instead, users can provide lightweight region inputs by drawing on a single frame, which the model then propagates across the video for spatial reasoning.

Combining these two aspects, SR-3D demonstrates robust spatial understanding from unconstrained video inputs without reliance on 3D scans or exhaustive annotations (Figure 1). These flexibilities open the door to a wide range of real-world applications, such as assisting robots in unstructured environments, analyzing large video collections, and supporting interactive spatial reasoning tasks.

# B COMPREHENSIVE LITERATURE REVIEW

**Region-level Vision-Language Models.** Region-level VLMs enhance fine-grained visual understanding by focusing on specific regions in images and videos. Early methods (Peng et al., 2024; Chen et al., 2023c;b; Wang et al., 2024e) represent regions as text using bounding box coordinates, making integration easy but relying on the language decoder for spatial reasoning. Others use visual markers like SoM (Yang et al., 2023), which overlay numbers and masks but alter image appearance and require rule-based placement. Another approach maps region features into LLM tokens using RoI-aligned features (Wang et al., 2024h;f; 2023a; Zhou et al., 2023; Zhang et al., 2023; Rasheed et al., 2024; Zhao et al., 2025), with RegionGPT (Guo et al., 2024) and Osprey (Yuan et al., 2024b) refining this by pooling pixel-level mask features for flexible region shapes. However, they struggle with resolution and aspect ratio constraints. In the video domain, various representations (Wang et al., 2024a; Yu et al., 2024; Fei et al., 2024; Ye et al., 2024; Heo et al., 2025) have been explored, but they mainly focus on tracking rather than multi-view spatial reasoning.

**Spatial Reasoning in Vision-Language Models.** Vision-language models have a strong visual understanding because they integrate the reasoning abilities of LLMs with powerful vision foundation models. Recently, there has been growing interest in equipping VLMs with spatial reasoning capabilities (Chen et al., 2024a; Ma et al., 2024a; Cai et al., 2025; Yuan et al., 2024a; Ma et al., 2024b; Tang et al., 2024; Song et al., 2024; Xu et al., 2025; Marsili et al., 2025; Liu et al., 2025b; Yang et al., 2025a; Liao et al., 2024). While most previous work has focused on spatial understanding from 2D images, multi-view spatial reasoning remains less explored. Recently, VSI-Bench (Yang et al., 2025b) was introduced as a testbed for evaluating models' 3D video-based spatial understanding. Our work extends this direction by proposing a unified 3D-aware architecture and representation that seamlessly supports both images and videos.

**3D Large Multimodal Models.** Our work also relates to recent advancements in 3D LMMs (Wang et al., 2023b; Man et al., 2024; Linghu et al., 2024; Hong et al., 2023; Fu et al., 2024a; Chen et al., 2024b; Huang et al., 2024c; Wang et al., 2025a; Huang et al., 2025; Wu et al., 2025). Various 3D representations have been explored to integrate position information into LLMs. 3D-LLM (Hong et al., 2023) and Scene-LLM (Fu et al., 2024a) use multi-view images with object segmentation masks to construct pixel-aligned point representations, while LL3DA (Chen et al., 2024b) directly employs a point cloud encoder to extract 3D scene features. LEO (Huang et al., 2024c) and Chat3D (Wang et al., 2023b) segment objects from the scene's point cloud and extract object features to represent the environment. These methods typically transform 3D scenes into voxel or point representations, but such approaches often limit the effectiveness of LLMs. Aligning these representations with LLMs requires vast amounts of data, which is challenging due to the scarcity of large-scale 3D datasets. Moreover, many of these methods rely on off-the-shelf 3d detection or segmentation models, which inherently constrain performance.

The most closely related works to ours are LLaVA-3D (Zhu et al., 2025) and Video-3D-LLM (Zheng et al., 2025), which also incorporate 3D position-aware features into 2D vision-language models. However, LLaVA-3D processes 3D and 2D data through separate pathways, while Video-3D-LLM fine-tunes 3D video data on a pre-trained video VLM. Both approaches risk overfitting 3D position encodings to specific 3D tasks. In contrast, our method adopts a unified architecture and 3D representation space for both image and video data, enabling better alignment and improving generalization across spatial understanding tasks.

## C   MORE QUANTITATIVE RESULTS ON 3D GENERAL BENCHMARKS

Following prior work, we report results using additional metrics for a more comprehensive evaluation. Table 10 presents results on Scan2Cap, Table 11 on ScanQA, and Table 12 on SQA3D. Apart from our method, all other results are from Video-3D-LLM (Zheng et al., 2025).

| | Cider ↑ | Bleu-4 ↑ | Meteor ↑ | Rouge ↑ |
|---|---|---|---|---|
| Scan2Cap (Chen et al., 2021) | 39.1 | 23.3 | 22.0 | 44.5 |
| 3DJCG (Cai et al., 2022) | 49.5 | 31.0 | 24.2 | 50.8 |
| D3Net (Chen et al., 2022) | 62.6 | 35.7 | 25.7 | 53.9 |
| 3D-VisTA (Zhu et al., 2023) | 66.9 | 34.0 | 27.1 | 54.3 |
| LL3DA (Chen et al., 2024b) | 65.2 | 36.8 | 26.0 | 55.1 |
| LEO (Huang et al., 2024c) | 68.4 | 36.9 | 27.7 | 57.8 |
| ChatScene (Huang et al., 2024b) | 77.2 | 36.3 | 28.0 | 58.1 |
| LLaVA-3D (Zhu et al., 2025) | 79.2 | 41.1 | 30.2 | 63.4 |
| Video-3D LLM (Zheng et al., 2025) | 83.8 | 42.4 | 28.9 | 62.3 |
| **SR-3D** | **97.9** | **44.7** | **31.5** | **67.3** |

Table 10: Full results on Scan2Cap (Chen et al., 2021) validation set.

| | EM | Bleu-1 ↑ | Bleu-2 ↑ | Bleu-3 ↑ | Bleu-4 ↑ | Rouge ↑ | Meteor ↑ | Cider ↑ |
|---|---|---|---|---|---|---|---|---|
| ScanQA (Azuma et al., 2022) | 21.1 | 30.2 | 20.4 | 15.1 | 10.1 | 33.3 | 13.1 | 64.9 |
| 3D-VisTA (Zhu et al., 2023) | 22.4 | – | – | – | 10.4 | 35.7 | 13.9 | 69.6 |
| Oryx-34B (Liu et al., 2025d) | – | 38.0 | 24.6 | – | – | 37.3 | 15.0 | 72.3 |
| LLaVA-Video-7B (Zhang et al., 2025) | – | 39.7 | 26.6 | 9.3 | 3.2 | 44.6 | 17.7 | 88.7 |
| 3D-LLM (Flamingo) (Hong et al., 2023) | 20.4 | 30.3 | 17.8 | 12.0 | 7.2 | 32.3 | 12.2 | 59.2 |
| 3D-LLM (BLIP2-flant5) (Hong et al., 2023) | 20.5 | 39.3 | 25.2 | 18.4 | 12.0 | 35.7 | 14.5 | 69.4 |
| Chat-3D (Wang et al., 2023b) | – | 29.1 | – | – | 6.4 | 28.5 | 11.9 | 53.2 |
| NaviLLM (Zheng et al., 2024) | 23.0 | – | – | – | 12.5 | 38.4 | 15.4 | 75.9 |
| LL3DA (Chen et al., 2024b) | – | – | – | – | 13.5 | 37.3 | 15.9 | 76.8 |
| Scene-LLM (Fu et al., 2024a) | 27.2 | 43.6 | 26.8 | 19.1 | 12.0 | 40.0 | 16.6 | 80.0 |
| LEO (Huang et al., 2024c) | – | – | – | – | 11.5 | 39.3 | 16.2 | 80.0 |
| Grounded 3D-LLM (Chen et al., 2024c) | – | – | – | – | 13.4 | – | – | 72.7 |
| ChatScene (Huang et al., 2024b) | 21.6 | 43.2 | 29.1 | 20.6 | 14.3 | 41.6 | 18.0 | 87.7 |
| LLaVA-3D (Zhu et al., 2025) | 27.0 | – | – | – | 14.5 | 50.1 | 20.7 | 91.7 |
| Video-3D LLM (Zhang et al., 2025) | 30.1 | 47.1 | 31.7 | 22.8 | 16.2 | 49.0 | 19.8 | 102.1 |
| **SR-3D** | **30.4** | **50.9** | **34.3** | **25.1** | **18.1** | **51.2** | **21.1** | **109.3** |

Table 11: Full results on ScanQA (Azuma et al., 2022) validation set.

| | What | Is | How | Can | Which | Others | Avg. |
|---|---|---|---|---|---|---|---|
| SQA3D (Ma et al., 2023) | 31.6 | 63.8 | 46.0 | 69.5 | 43.9 | 45.3 | 46.6 |
| 3D-VisTA (Zhu et al., 2023) | 34.8 | 63.3 | 45.4 | 69.8 | 47.2 | 48.1 | 48.5 |
| LLaVA-Video(Zhang et al., 2025) | 42.7 | 56.3 | 47.5 | 55.3 | 50.1 | 47.2 | 48.5 |
| Scene-LLM (Fu et al., 2024a) | 40.9 | 69.1 | 45.0 | 70.8 | 47.2 | 52.3 | 54.2 |
| LEO (Huang et al., 2024c) | – | – | – | – | – | – | 50.0 |
| ChatScene (Huang et al., 2024b) | 45.4 | 67.0 | 52.0 | 69.5 | 49.9 | 55.0 | 54.6 |
| LLaVA-3D (Zhu et al., 2025) | – | – | – | – | – | – | 55.6 |
| Video-3D LLM (Zheng et al., 2025) | 51.1 | 72.4 | 55.5 | 69.8 | 51.3 | 56.0 | 58.6 |
| **SR-3D** | **55.0** | **76.4** | **59.8** | **71.6** | **54.7** | **61.1** | **62.2** |

Table 12: Full results on SQA3D (Ma et al., 2023) testing set.

## D   MORE QUALITATIVE RESULTS ON VSI-BENCH

We report additional visual results on VSI-Bench, primarily using scenes from ScanNet++. ScanNet++ is not included in EmbodiedScan's annotations, making it a distinct and challenging dataset for evaluation. Compared to ScanNet, ScanNet++ offers higher fidelity and greater diversity in indoor environments. Moreover, its 3D annotations are only coarsely aligned to match walls and floors to the axis. Despite these challenges, as shown in Figure 6, our method demonstrates superior capabilities in determining relative direction, highlighting its robustness in real-world tasks.

## E   MORE ABLATION STUDY ON TILE-AND-STITCH

We conduct an ablation study to isolate the effect of the tile-and-stitch module. Using the same NVILA-Lite-8B checkpoint and training on identical data (V3Det + VG, etc., consistent with RegionGPT), we evaluate COCO classification with and without tile-and-stitch. The results demonstrate consistent improvements, particularly on small objects, indicating that tile-and-stitch increases the effective receptive resolution and directly enhances recognition performance.

| Method | $AP_{Small}$ | $AP_{Medium}$ | $AP_{Large}$ | mAP | Acc |
|---|---|---|---|---|---|
| Baseline | 50.4 | 74.0 | 80.5 | 66.2 | 79.8 |
| + tile-and-stitch | 67.5 | 79.6 | 80.3 | 76.3 | 86.9 |
| $\Delta$ | +17.1 | +5.6 | −0.2 | +10.1 | +7.1 |

Table 13: Ablation study isolating the effect of the tile-and-stitch module on COCO classification.

## F   MORE ABLATION STUDY ON ALTERNATIVE 3D REPRESENTATIONS

We compare four settings: no 3D representation, our 3D positional encoding (3D PE) point representation, VGGT features, and Pi3 features. All forms of 3D information improve performance, with 3D PE and Pi3 providing the strongest gains. While foundation-model features (e.g., VGGT and Pi3) are effective, 3D PE is more flexible because it is not tied to a specific pretrained backbone. When a 3D foundation model is updated or replaced, feature-based approaches typically require full retraining, whereas 3D PE can directly utilize pointmaps from arbitrary sources. Furthermore, some foundation models (e.g., VGGT) aggregate features from multiple layers, resulting in very high-dimensional embeddings that may be more difficult for the model to fuse effectively.

| Method | ScanQA | SQA3D | Scan2Cap |
|---|---|---|---|
| Baseline (no 3D repr.) | 28.1 | 59.0 | 89.1 |
| + VGGT | 28.6 | 59.2 | 95.2 |
| + $\pi^3$ | 29.0 | 59.2 | 98.8 |
| + 3D PE | **29.1** | **59.5** | 97.3 |

Table 14: Ablation study on alternative 3D representations.

## G   MORE ABLATION STUDY ON PRE-TRAINING AND POSITIONAL ENCODING

We present the complete ablation study results on 2D single-view pre-training and 3D positional encoding without pre-training, evaluating their influence on model performance. The detailed results are shown in Table 15 and Table 18, respectively.

Overall, the fully-trained model consistently outperforms baseline models on 3D general QA benchmarks, demonstrating the benefits of leveraging both 2D and 3D spatial information. However, in the 3D spatial-focused dataset, we observe a slight drop in the Wide and Big category, likely due to differences in how width is defined in 2D versus 3D, as discussed in the main paper.

Additionally, we find that removing pre-training leads to a substantial drop in performance for more complex reasoning tasks, particularly in the multi-choice complex category, where the model struggles without prior exposure to large-scale 2D pre-training. These results highlight the importance of both spatial-aware representation learning and strong pre-training strategies in enhancing 3D reasoning capabilities.

## H   STATISTICS OF SR-3D-BENCH

Our benchmark follows template designs from prior works on spatial reasoning in vision-language models, including SpatialRGPT (Cheng et al., 2024) and SpatialVLM (Chen et al., 2024a). To further increase the complexity and diversity of spatial reasoning tasks, we incorporate situated annotations from the EmbodiedScan (Wang et al., 2024d) dataset, ensuring a more realistic and challenging evaluation setting. Specifically, our dataset includes a range of spatial relationships,

| | | Scan2Cap | | | | ScanQA | | | | SQA3D | |
|---|---|---|---|---|---|---|---|---|---|---|---|
| PE | PT | Bleu-4 ↑ | Rouge↑ | Cider↑ | Meteor↑ | Bleu-4 ↑ | Rouge↑ | Cider↑ | Meteor↑ | EM ↑ | EM ↑ |
| | | 44.2 | 67.3 | 92.9 | 31.1 | 16.0 | 48.9 | 101.3 | 19.8 | 28.8 | 58.6 |
| ✔ | | 44.0 | 67.3 | 92.7 | 31.0 | 17.4 | 48.8 | 102.9 | 20.0 | 29.1 | 59.1 |
| ✔ | ✔ | **44.7** | **67.3** | **97.9** | **31.5** | **18.1** | **51.2** | **109.3** | **21.2** | **30.4** | **62.2** |

Table 15: Ablation study full results on Scan2Cap, ScanQA, and SQA3D benchmarks.

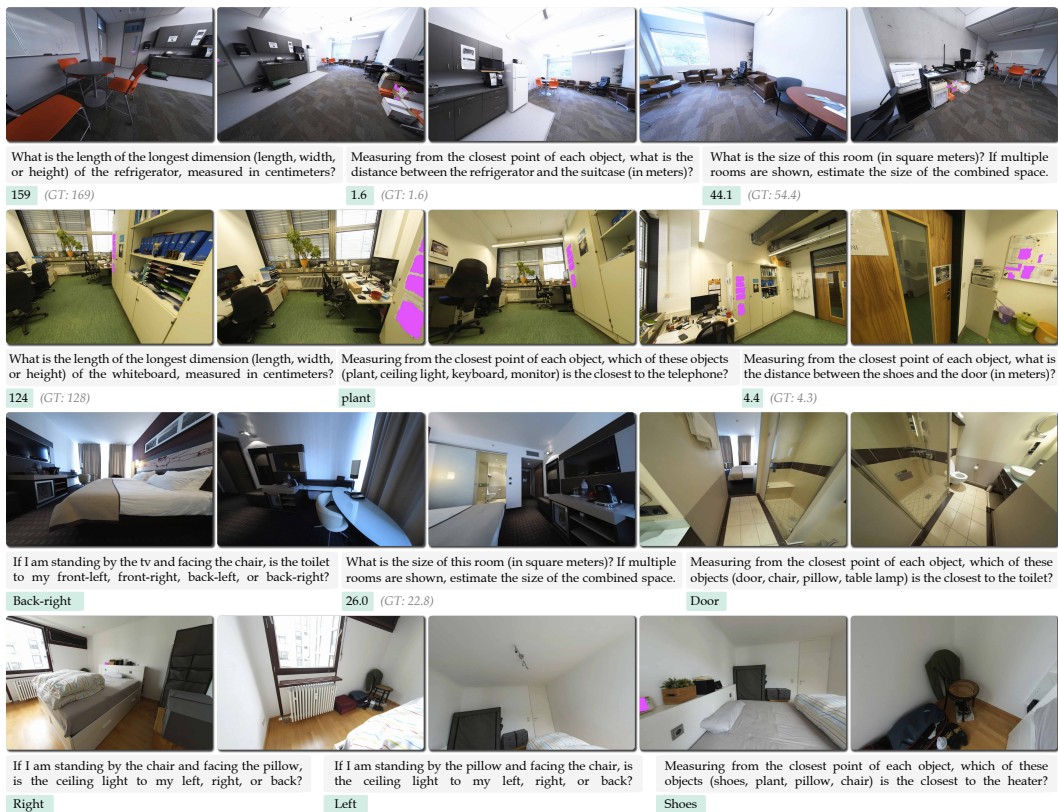

Figure 6: More results on VSI-Bench (Yang et al., 2025b). We highlight SR-3D 's outputs and include ground-truth values for numerical answers.

| Category | Thin-Wide | Tall-Short | Big-Small | Multi-Simple | Multi-Complex | Width Data | Distance Data | Height Data | Total Length |
|---|---|---|---|---|---|---|---|---|---|
| Count | 219 | 231 | 231 | 117 | 500 | 496 | 242 | 464 | 2500 |

Table 16: Statistical analysis of our SR-3D-Bench, showing the distribution of different spatial attributes.

from basic geometric comparisons such as thin-wide, tall-short, and big-small, to more complex multi-object interactions categorized as multi-simple and multi-complex. Additionally, we introduce explicit width, distance, and height annotations to facilitate fine-grained spatial understanding. With a total of 2,500 samples, our benchmark provides a comprehensive evaluation for assessing the region-level spatial reasoning capabilities of vision-language models in realistic scenarios.

# I IMPLEMENTATION DETAILS OF SR-3D

We use vision encoder from the SigLIP module inside PaliGemma (Beyer et al., 2024) as our visual backbone with an input size of 448 and a patch size of 14, paired with a Qwen-2-7B (Bai et al., 2023) LLM backbone. For training the foundational 2D VLM, we follow prior work and set the maximum tiles per image to 12. For the multi-view VLM, we use a frame size of 32 with a uniform sampling strategy to ensure a fair comparison with previous methods. For training the 2D VLM, we adopt a learning rate of 5e-5 with cosine decay and gradient clipping enabled. The same hyperparameters are used for fine-tuning the 3D VLM, except for a reduced batch size due to the increased token length.

| | | |
|---|---|---|
| ***2D Data*** | | |
| | Hybrid | ShareGPT4V-SFT, Molmo, The Cauldron, Cambrian, LLaVA-OneVision |
| | Captioning | MSR-VTT, Image Paragraph Captioning, ShareGPT4V-100K |
| | Reasoning | CLEVR, NLVR, VisualMRC |
| | Document | DocVQA, UniChart-SFT, ChartQA |
| | OCR | TextCaps, OCRVQA, ST-VQA, POIE, SORIE, SynthDoG-en, TextOCR-GPT4V, ArxivQA, LLaVAR |
| | General VQA | ScienceQA, VQAv2, ViQuAE, Visual Dialog, GQA, Geo170K, LRV-Instruction, RefCOCO, GeoQA, OK-VQA, TabMVP, EstVQA |
| | Diagram & Dialogue | DVQA, AI2D, Shikra, UniMM-Chat |
| | Instruction | LRV-Instruction, SVIT, MMC-Instruction, MM-Instruction |
| | Text-only | FLAN-1M, MathInstruct, Dolly, GSM8K-ScRel-SFT |
| | Knowledge | WordART, WIT, STEM-QA |
| | Medical | PathVQA, Slake, MedVQA |
| | Region | RegionGPT |
| | Spatial | SpatialRGPT |
| ***3D Data*** | | |
| | General | ScanQA, SQA3D, Scan2Cap |
| | Spatial | EmbodiedScan |

Table 17: Data recipe for training 2D foundational VLM and 3D fine-tuning.

| | | 3D Region | | | | | | 3D Global | | | |
|---|---|---|---|---|---|---|---|---|---|---|---|
| PE | PT | Wide | Tall | Big | M. Sim. | M. Cpx. | Avg. | Width | Height | Dist. | Avg. |
| | | **77.6** | 80.5 | **82.6** | 71.7 | 55.8 | 73.6 | 85.8 | 84.4 | 53.7 | 74.4 |
| ✔ | | **77.6** | **83.1** | 80.5 | 70.9 | 59.0 | 74.2 | 85.5 | 85.7 | 60.3 | 77.2 |
| ✔ | ✔ | 76.3 | **83.1** | 81.8 | **80.3** | **76.0** | **79.5** | **87.7** | **87.3** | **74.8** | **83.3** |

Table 18: Ablation study full results on 3D region and 3D global tasks.

The data recipe for both training stages is detailed in Table 17. We train on a subset of 2D data, excluding spatial and region-related datasets, to preserve the original vision-language capabilities while incorporating a diverse source.

## J COMPUTATIONAL AND MEMORY OVERHEAD

The region extractor introduces only lightweight intermediate tensors of size $R \times T$, where $R$ denotes the number of region masks and $T$ is the post-tiling token count. Consequently, its memory and computational overhead are negligible relative to the backbone; nearly all additional cost stems from the increased number of visual tokens introduced by tiling.

Implementation-wise, the tile-then-stitch extractor is efficient. It operates directly on the token grid of length $T$ (rather than full-resolution pixels), batches all region masks per sample into a single operation, and performs pooling using a single GPU-efficient `einsum` over concatenated features, thereby avoiding per-region encoder passes. Empirically, the region extraction step consistently takes less than 1 ms on a single A100 GPU and contributes only a small fraction of the end-to-end inference time.

| Setting | Total Inference Time (s) | Region Extract Time (s) |
|---|---|---|
| w/o tiling | 0.221 | 0.000375 |
| w/ tiling ($512 \times 512 \to 2 \times 2 + 1$ tiles) | 0.269 | 0.000529 |
| w/ tiling ($1024 \times 1024 \to 3 \times 3 + 1$ tiles) | 0.459 | 0.000757 |

Table 19: Inference time comparison under different tiling settings.

## K  More Discussions

**Computational Efficiency Compared to LEO.** Object-centric methods such as LEO are more token-efficient, typically using fewer than 100 tokens per scene, whereas our image-based approach uses $196 \times 32$ tokens. However, these methods rely on external components such as Mask3D, PointNet++, and Spatial Transformers, which can introduce additional system complexity and may discard fine-grained visual details. In contrast, SR-3D provides a more end-to-end solution. Furthermore, object-centric approaches often discard temporal information, whereas our image-based design preserves frame ordering and naturally supports tasks requiring appearance-order reasoning, such as those evaluated in VSI-Bench.

**General Video Understanding Capabilities.** SR-3D is built on NVILA-Lite, an image-based VLM that has never been trained on videos. To answer this question, we conduct an additional co-training experiment using multi-view data together with a subset of LLaVA-Video's training data. The comparison with LLaVA-3D is shown in Table 20.

| Method | ScanQA | SQA3D | Scan2Cap | VSI-Bench | VideoMME | MVBench |
|---|---|---|---|---|---|---|
| LLaVA-3D | 103.1 | 60.1 | 84.1 | — | 62.8 | 58.1 |
| SR-3D | **105.1** | 59.2 | **97.1** | 66.2 | 58.1 | **60.3** |

Table 20: Comparison between LLaVA-3D and SR-3D across multiple benchmarks.

**Evaluation on MMScan.** MMScan Lyu et al. (2024) provides comprehensive 3D evaluation spanning object- to room-level understanding. We did not include comparisons on MMScan due to computational cost constraints. We were unable to reproduce the "29k validation samples" referenced in LLaVA-3D. Evaluating the full 318k split would be extremely expensive, as it is effectively equivalent to testing on 318k videos. We leave thorough evaluation on MMScan for future work.

## L  Limitations

**Orientations.** Although our method shows promising results, it remains challenging for current vision-language models to accurately perceive and interpret spatial questions related to object orientation. This challenge arises due to the difficulty of scaling up data. We leave this as future work.

**Dynamic Videos.** Our method is designed for multi-view static data, whereas real-world scenarios often involve dynamic environments. Incorporating positional embeddings to handle both static and dynamic inputs is non-trivial. Future work should explore methods to address this limitation.

**OCR Tasks.** In the main paper of Table 1, we report the performance of our 2D foundation model on general benchmarks. While our model maintains comparable performance to the base model, demonstrating improved spatial understanding without significant trade-offs, we observe a consistent slight drop in OCR-related tasks. A potential solution is to incorporate more OCR-related tasks into the training data pipeline.

**Unified Checkpoint.** While our unified architecture and representation provide a foundation for both single- and multi-view 3D-aware VLMs, we leave it to future work to investigate how to effectively combine the two models. This could be achieved either by introducing an agentic flow between single- and multi-view models or by directly training a single model across both settings, which may further improve generalization and efficiency.

## M  Use of LLM

To improve the clarity and presentation of this manuscript, we used large language models for minor editorial suggestions on grammar and sentence structure. The core scientific ideas, experimental work, and original text were authored exclusively. We critically evaluated every change proposed by the LLM to guarantee that the final manuscript is a faithful and accurate representation of our research and findings.

