# OpenReview forum: "3D Aware Region Prompted Vision Language Model"
_ICLR.cc/2026/Conference — ICLR 2026 Poster_

### Official Review · Reviewer_TQXC · 2025-10-27

**Soundness:** 3
**Presentation:** 4
**Contribution:** 3
**Rating:** 6
**Confidence:** 4

**Summary:**

This paper introduces a method to unify single-view and multi-view tasks within a single vision-language model. To achieve this, the authors propose a novel **dynamic tiling-based region extractor** that produces generalizable embeddings across single- and multi-view settings. With the proposed approach, the **SR-3D** model achieves state-of-the-art (SOTA) performance on general multimodal and 3D understanding benchmarks.

**Strengths:**

1. The idea of extracting canonical positional features shared across single- and multi-view inputs is simple yet effective.
2. The proposed method demonstrates strong generalization from 2D multimodal to 3D multimodal scenarios. Its impressive zero-shot performance on VSI-Bench highlights this capability.
3. The method leverages diverse 3D representations and can generalize to pseudo point maps derived from 3D foundation models, showing its ability to integrate SOTA VLMs with 3D foundation models.

**Weaknesses:**

1. For multi-view inputs, the SR-3D model requires 32 frames, whereas other 3D-input VLMs such as LEO use only around 60 object-centric tokens. This substantially increases sequence length and may affect training and inference efficiency, which should be discussed.
2. Related works such as **LLaVA-3D** also evaluate video understanding on benchmarks like **MVBench [1]** and **VideoMME [2]**. Although the proposed method exhibits strong spatial understanding in multi-view settings, it remains unclear whether the enhanced 3D understanding compromises its general video understanding ability.
3. The 3D evaluation is not comprehensive enough. It seems that the 2D and 3D evaluations were conducted using different models. The proposed **SR-3D-Bench** focuses primarily on region-level spatial scene understanding, and the baselines are limited to video-based VLMs. Recent benchmarks such as **MMScan [3]** offer more comprehensive 3D evaluations from object- to room-level, and results on them are recommended for inclusion.

[1] [[2311.17005\] MVBench: A Comprehensive Multi-modal Video Understanding Benchmark](https://arxiv.org/abs/2311.17005)

[2] [[2405.21075\] Video-MME: The First-Ever Comprehensive Evaluation Benchmark of Multi-modal LLMs in Video Analysis](https://arxiv.org/abs/2405.21075)

[3] [[2406.09401\] MMScan: A Multi-Modal 3D Scene Dataset with Hierarchical Grounded Language Annotations](https://arxiv.org/abs/2406.09401)

**Questions:**

1. **Use of masks or bounding boxes during evaluation:** In real-world scenarios, obtaining ground-truth object masks or bounding boxes is challenging. Does the method rely on GT instance masks or boxes during evaluation, or are these extracted using 3D foundation models such as **Mask3D [4]**? Additionally, how sensitive are the results to mask quality?
2. **Integration of Paligemma (L1126):** How is **Paligemma** used as the visual backbone, given that it is a VLM? Furthermore, why was **Qwen-2-7B** selected as the LLM backbone? Would more recent models such as **Qwen-2.5-7B** yield better results?
3. **Checkpoint consistency:** Are the checkpoints used for 2D and 3D evaluations identical? If not, does the 3D model still retain its 2D understanding ability?
4. **Base model selection:** Why was **NVILA-Lite-8B** chosen as the base model? Does the 2D model use similar pretraining data or a comparable data recipe?

[4] [[2210.03105\] Mask3D: Mask Transformer for 3D Semantic Instance Segmentation](https://arxiv.org/abs/2210.03105)

---

> ### Author Response · Authors · 2025-11-24
> **Response to Reviewer TQXC (1/2)**
>
> We thank the reviewer for their valuable feedback. We address your comments as follows.
> ***
> >**W1:** *SR-3D has a longer sequence length compared to LEO and may impact training and inference efficiency.*
>
> **A1:** Thank you for the suggestion; we will add this discussion to the revised paper. Object-centric methods like LEO are indeed more efficient, using fewer than 100 tokens per scene, while our image-based approach uses 196 × 32 tokens. However, these methods depend on external components such as Mask3D, PointNet++, and Spatial Transformers, which may lose visual details and add complexity. SR-3D provides a more end-to-end solution. In addition, object-centric approaches typically drop temporal information, while our image-based design preserves frame order and supports tasks like appearance-order reasoning in VSI-Bench.
> ***
>
> >**W2:** *LLaVA-3D also evaluates video understanding on benchmarks like MVBench and VideoMME. Does enhanced 3D understanding compromises its general video understanding ability?*
>
> **A2:** LLaVA-3D is built on LLaVA-Video and inherits a video-based VLM trained extensively on video data. In contrast, SR-3D is built on NVILA-Lite, an image-based VLM that has never been trained on videos. To address the reviewer’s question, we conduct an additional co-training experiment using multi-view data together with a subset of LLaVA-Video’s training data. The comparison with LLaVA-3D is shown below.
>
> \begin{array}{lcccccc}
> \hline
> \textbf{Method} & \textbf{ScanQA} & \textbf{SQA3D} & \textbf{Scan2Cap} & \textbf{VSI-Bench} & \textbf{VideoMME} & \textbf{MVBench} \newline
> \hline
> \text{LLaVA-3D} & 103.1 & 60.1 & 84.1 & - & 62.8 & 58.1 \newline
> \text{SR-3D (Ours)} & 105.1 & 59.2 & 97.1 & 66.2 & 58.1 & 60.3
> \newline
> \hline
> \end{array}
> ***
>
> >**W3:** *The 3D evaluation is not comprehensive enough. SR-3D baselines are limited to video-based VLMs. Recommend including MMSCan.*
>
> **A3:**
> Thank you for the suggestion to include MMScan. We will add a discussion of MMScan in the revised version. However, we were not able to reproduce the “29k validation samples” referenced in LLaVA-3D. The MMScan paper reports 297k samples, and the official GitHub release contains 318k. Evaluating on the full 318k split would be extremely costly, as it is equivalent to testing on 318k videos, and the exact split used in prior work is not clearly documented, making direct comparison difficult.
>
> The three benchmarks we evaluate on—ScanQA, Scan2Cap, and SQA—are well-established standards in 3D vision-language research. Together, they span from object-level to scene-level spatial understanding. The baselines we compare against also cover all major 3D methods used in MMScan (3D-LLM, Chat3D-v2, LL3DA, LEO, LLaVA-3D), ensuring that our evaluation is comprehensive and aligned with the key models in the field.
>
> ***

---

> ### Author Response · Authors · 2025-11-24
> **Response to Reviewer TQXC (2/2)**
>
> >**Q1:** *Use of masks or bounding boxes during evaluation.*
>
> **A4:** No, SR-3D does not rely on ground-truth masks or boxes. The model uses image–region inputs rather than explicit 3D coordinates. During evaluation, we obtain 2D masks using SAM, which aligns with real-world usage where a user can simply provide a 2D box or point and let SAM generate the mask.
>
> The model is also not sensitive to mask quality. As noted in L234, we apply several mask augmentations during training, such as converting masks to boxes, randomly dropping frames to mimic single-frame annotations, and using partially obscured or tiny masks. These augmentations make the region embeddings robust to imperfect or noisy boundaries.
> ***
>
> >**Q2:** *Integration of Paligemma (L1126)*
>
> **A5:** We use Paligemma and Qwen-2-7B because they are part of the base model we adopt, NVILA-Lite-8B. NVILA-Lite-8B initializes its vision encoder from the SigLIP module inside Paligemma, not the full VLM. We will make this clearer in the revision. While newer LLMs such as Qwen-2.5-7B may offer further improvements, our model already achieves state-of-the-art results even with Qwen-2-7B.
> ***
>
> >**Q3:** *Checkpoint consistency.*
>
> **A6:** The 2D and 3D evaluations use separate checkpoints. While training a single unified checkpoint is possible (L1147), the main obstacle is computational rather than methodological. Co-training both 2D and 3D data would require roughly 1/4 of the effective batch size, leading to over 2500 GPU-days, and it also requires careful data balancing, so a simple one-shot run is unlikely to work. However, our earlier 3D+video co-training experiment (A2) demonstrates that joint training is feasible: the model retains strong 2D ability while gaining 3D understanding. This supports our claim that a unified checkpoint is achievable when resources permit.
> ***
>
> >**Q4:** *Base model selection.*
>
> **A7:** We adopt  NVILA-Lite-8B because it is efficient for both training and deployment (as described in the original NVILA paper), which aligns with our computational constraints. In the 2D pretraining stage, we use only about half of the original NVILA data recipe, yet still achieve comparable performance on standard VLM benchmarks (Table 1).

---

> > ### Comment · Reviewer_TQXC · 2025-11-26
> >
> > Thank you for the clarification.
> > I will keep my score unchanged but increase my confidence.

---

### Official Review · Reviewer_2EZD · 2025-10-29

**Soundness:** 2
**Presentation:** 3
**Contribution:** 2
**Rating:** 4
**Confidence:** 4

**Summary:**

This paper presents a 3D multi-view spatial reasoning framework based on 2D VLMs with region-level prompts. The core insight is that spatial understanding fundamentally relies on cross-frame consistency of the same object. To this end, the authors introduce a unified 3D positional encoding to enrich the visual features of powerful 2D VLMs, thereby leveraging their strong priors while mitigating the scarcity of 3D data. In addition, they adopt a tile-and-stitch strategy for both visual features and masks to ensure fine-grained region-level feature extraction. Experiments demonstrate that the proposed method achieves notable improvements on various spatial reasoning and scene understanding benchmarks.

**Strengths:**

1. The motivation is clear. The paper identifies that the key to spatial understanding and reasoning lies in maintaining cross-frame consistency for the same object, and accordingly designs a 3D positional encoding module, complemented by a tile-and-stitch strategy.
2. The training pipeline is well-designed, effectively preserving the strong priors of the underlying VLM.
3. The framework can operate with estimated depth in the absence of ground-truth depth and supports flexible region prompting, showing promising practical potential.
4. Extensive experiments are conducted, and the figures and visualizations are well-presented and visually appealing.

**Weaknesses:**

1. The idea of 3D positional encoding has been widely explored in prior 3D modeling and vision literature, and the proposed design does not appear to introduce substantial contribution compared to existing approaches.
2. As shown in Table 8 (ablation study), the 3D positional encoding contributes only marginal improvement, while most performance gains come from single-view pretraining. However, the paper’s main claim centers on the role of 3D positional encoding in spatial understanding and reasoning.
3. While the experimental results are solid overall, the core idea lacks strong conceptual contribution, and the experiments do not convincingly demonstrate the effectiveness of 3D positional encoding in enhancing spatial understanding and reasoning.

**Questions:**

1. The zero-shot results in Table 7 and the visualizations in Table 5 suggest that SR-3D exhibits strong sensitivity to 3D distances. Is this metric-scale capability primarily attributable to the 3D positional encoding design? Which specific components contribute to it? Please provide theoretical justification and supporting experiments.
2. In the ablation study (Table 8), the gains from 3D positional encoding are limited, while most improvements come from single-view pretraining. Since 3D PE is a key claimed contribution, please explain this discrepancy.
3. The tile-and-stitch component lacks detailed ablation studies. Please include analyses isolating its effect.
4. When depth ground truth is unavailable during training and only estimated depth is used, how much does performance degrade? In particular, could depth bias be amplified after transforming to the unified 3D coordinate system for PE, thereby harming cross-frame alignment and metric measurement?
5. Beyond 3D PE, have you explored alternative 3D representations (e.g., explicit depth features, VGGT or Depth Anything features)?
6. I’m open to revising my score after the rebuttal.

---

> ### Author Response · Authors · 2025-11-24
> **Response to Reviewer 2EZD (1/2)**
>
> We thank the reviewer for their valuable feedback. We address your comments as follows.
> ***
> >**W1:** *3D PE has been widely explored in prior work and does not introduce substantial contribution compared to existing approaches.*
>
> **A1:** Prior works do explore 3D PE, but their effectiveness has been very limited. For example, LLaVA-3D reports only +0.4 on ScanQA and +0.3 on SQA3D, and Video3dLLM even reports –0.41 on ScanQA. These results show that simply injecting 3D coordinates does not meaningfully improve VLM spatial reasoning.
>
> Our contribution is not the use of 3D PE alone, but a **unified camera-aligned representation** that allows 3D PE to work together with **million-scale single-view depth pretraining**. This design makes 3D PE effective across both single-view and multi-view tasks. The strong performance gains we observe are possible only because of this unified formulation, which has not been demonstrated in prior 3D-aware VLMs.
> ***
> >**W2:** *3D positional encoding contributes only marginal improvement, while most performance gains come from single-view pretraining.*
>
> **A2:** Table 8 directly supports our contribution. Using the same pretraining data, adding 3D PE provides nearly +3 improvement on SQA3D (and similar gains on other tasks) over training without PE. In contrast, using 3D PE without this pretraining gives less benefit. This shows two key points:
> 1. **3D PE alone is not enough**, which aligns with prior findings.
> 2. **3D PE becomes most effective only when paired with dense 2.5D pretraining**, which our unified representation design uniquely enables.
>
> Therefore, the observed gains are genuine improvements rather than artifacts.
> ***
> **W3:** *The core idea lacks strong conceptual contribution and the experiments do not convincingly demonstrate the effectiveness of 3D PE.*
>
> **A3:** Our work does not claim that 3D PE itself is novel or inherently powerful. Instead, the core contribution is the unified representation that makes 3D PE effective by enabling large-scale single-view geometric pretraining in a consistent camera-aligned space. Our ablation results directly support this: naively adding 3D PE provides only marginal gains, which aligns with prior work and reinforces our point that 3D PE alone is insufficient. The substantial improvements come only when 3D PE is combined with our unified design and million-scale 2.5D pretraining. SR-3D is the first to demonstrate a scalable formulation where 3D PE contributes meaningful gains across diverse 3D spatial tasks.
> ***

---

> ### Author Response · Authors · 2025-11-24
> **Response to Reviewer 2EZD (2/2)**
>
> >**Q1:** *Is the metric-scale capability primarily attributable to the 3D PE design?*
>
> **A4:** We conduct further analysis to support the zero-shot 3D distance results shown in Table 7. The base model, NVILA-Lite, is not capable of metric-scale distance prediction on its own. The metric understanding comes from two components in our design: (1) the pretraining data derived from SpatialRGPT, which provides metric distance supervision, and (2) the 3D positional encoding aligned in our unified camera space. Empirically, when we pretrain on the same data without 3D PE, the score is **41.7**, whereas with 3D PE it increases to **68.5**. This large gap shows that the unified 3D PE design plays a key role in grounding metric distances, while the pretraining data provides the necessary scale information. Together, these components enable SR-3D’s strong metric-scale reasoning ability.
> ***
>
> >**Q2:** *Table 8, gains from 3D positional encoding are limited, while most improvements come from single-view pretraining*
>
> **A5:** Please see our above response to W1, W2, and W3.
> ***
>
> >**Q3:** *The tile-and-stitch component lacks detailed ablation studies*
>
> **A6:** We include an ablation isolating the effect of the tile-and-stitch module. Starting from the same NVILA-Lite-8B checkpoint and training on identical data (V3Det + VG, etc., as used in RegionGPT), we evaluate COCO classification with and without tile-and-stitch. Results below show clear gains, especially on small objects, confirming that tile-and-stitch provides higher effective receptive resolution and directly improves recognition.
>
> \begin{array}{lccccc}
> \hline
> \bf{Method} & \bf{AP} _{Small}& \bf{AP} _{Medium}& \bf{AP} _{Large}& \bf{mAP} & \bf{Acc} \newline
> \hline
> \text{Baseline} & 50.4 & 74.0 & 80.5 & 66.2 & 79.8 \newline
> \text{+ tile-and-stitch} & 67.5 & 79.6 & 80.3 & 76.3 & 86.9 \newline
> \hline
> \Delta & +17.1 & +5.6 & -0.2 & +10.1 & +7.1 \newline
> \hline
> \end{array}
> ***
>
> >**Q4:** *When estimated depth is used for training, how much does performance degrade? Could depth bias be amplified after transforming to the unified 3D coordinate system for PE?*
>
> **A7:** Our method does not require pixel-accurate depth. The 3D positional encoding operates at the token level and only needs coarse geometric structure, which estimated depth can reliably provide. The unified 3D coordinate system is normalized, so it is not sensitive to absolute depth scale and does not amplify depth bias. To verify this, we retrained the model using Pi3-estimated pointmaps and evaluated on VSI-Bench with both ground-truth and estimated depth. The scores differ only slightly, indicating that the model is robust to moderate depth estimation noise.
>
> \begin{array}{lcc}
> \hline
> \textbf{Training} & \textbf{Testing} & \textbf{Score} \newline
> \hline
> \pi^{3} & \pi^{3} & 62.9 \newline
> \pi^{3} & \text{GT} & 63.3 \newline
> \text{GT}  & \text{GT} & 62.9 \newline
> \hline
> \end{array}
> ***
>
> >**Q5:** *Explore alternative 3D representations.*
>
> **A8:** Following the request, we compared four settings: no 3D representation, our 3D PE point representation, VGGT features, and Pi3 features. All forms of 3D information help, with 3D PE and Pi3 providing the strongest gains. While foundation-model features (VGGT, Pi3) work well, 3D PE is more flexible because it is not tied to a specific pretrained backbone. If a 3D foundation model is updated or retrained, feature-based approaches would require full retraining, whereas 3D PE can directly use pointmaps from any source. In addition, some foundation models (e.g., VGGT) combine features from multiple layers, creating very high-dimensional embeddings that may be harder for the model to fuse effectively. We will include this discussion in our revised paper.
>
> \begin{array}{lccc}
> \hline
> \textbf{Method} & \textbf{ScanQA} & \textbf{SQA3D} & \textbf{Scan2Cap} \newline
> \hline
> \text{Baseline (no 3D repr.)} & 28.1 & 59.0 & 89.1 \newline
> \text{+ VGGT} & 28.6 & 59.2 & 95.2 \newline
> \text{+ }\pi^{3} & 29.0 & 59.2 & 98.8 \newline
> \text{+ 3D PE} & \bf{29.1} & \bf{59.5}& \underline{97.3} \newline
> \hline
> \end{array}
> ***

---

> > ### Comment · Reviewer_2EZD · 2025-11-26
> > **Post Rebuttal**
> >
> > Thank you for the clarifications and the additional experiments. My main concerns are addressed, and I will increase my score to 6.

---

### Official Review · Reviewer_jRLg · 2025-10-31

**Soundness:** 2
**Presentation:** 3
**Contribution:** 2
**Rating:** 6
**Confidence:** 4

**Summary:**

The paper introduces SR-3D, a 3D-aware VLM that unifies 2D and 3D spatial understanding through a shared visual token space. It bridges single-view (2D) and multi-view (3D) representations by enriching 2D features with 3D positional embeddings derived from depth maps, aiming to enable spatial reasoning across frames even when objects do not co-occur.
The model is trained first on large-scale 2D datasets for strong visual-language priors, then fine-tuned on multi-view 3D datasets (e.g., ScanQA, SQA3D, Scan2Cap). The authors also evaluate the performance on 3D spatial reasoning, region-level understanding, and video-based spatial question answering benchmarks such as BLINKDepth, ScanQA, and VSI-Bench.

**Strengths:**

1. Well-designed figures: The figures are clear, well-organized, and effectively illustrate both the overall pipeline and qualitative examples.
2. Comprehensive evaluation: The paper presents extensive experiments across 2D, 3D, and video-based spatial reasoning benchmarks.
3. Practical and flexible design: The proposed SR-3D framework supports flexible region prompting (e.g., bounding boxes, masks, or 3D annotations)

**Weaknesses:**

1. The claim in the contribution "We introduce SR-3D, the first 3D-aware vision-language model that unifies representations for both single-view and multi-view tasks." might be a bit overstated, while not exactly the same but existing works have already explore similar idea such as LLaVA-3D, 3D-LLM etc

2. The authors emphasize the unified representation between single- and multi-view images. While this idea is conceptually sound, it may be difficult to directly demonstrate that the model indeed develops such a capability, or that the improvements reported in the tables stem from this specific design. It would be valuable if the authors could provide additional evidence or analysis to better support this claim.

**Questions:**

Please address the weakness mentioned above.

---

> ### Author Response · Authors · 2025-11-24
> **Response to Reviewer jRLg (1/1)**
>
> We thank the reviewer for their valuable feedback. We address your comments as follows.
> ***
> >**W1:** *Contribution “the first 3D-aware vision-language model that unifies representations for both single-view and multi-view tasks” overstated. Existing works have already explored similar idea.*
>
> **A1:** While LLaVA-3D and Video3dLLM also explore adding 3D positional cues, their gains are very limited (LLaVA-3D: +0.4 on ScanQA, +0.3 on SQA3D, and Video3dLLM even –0.41 on ScanQA), showing that **simply injecting 3D PE does not reliably improve VLM performance**.
>
> Our contribution goes beyond using 3D PE. We provide a **unified design that aligns camera space and enables million-scale RGB-relative-depth pretraining**, which turns 3D PE into meaningful improvements across both single-view and multi-view tasks. Our model achieves much stronger results than prior 3D-aware VLMs, demonstrating that SR-3D is the first to make this approach work effectively in practice. We will adjust our key claim to clarify this distinction.
> ***
>
> >**W2:** *Do the improvements reported in the tables stem from the unified representation?*
>
> **A2:** Yes, and Table 8 provides clear evidence. Pretraining with 3D PE gives nearly +3 improvement on SQA3D (and similar gains on other tasks) compared to (1) pretraining+finetuning without PE (2) 3D PE without pretraining. **Pretraining with PE is only possible with our unified representation design**, so the consistent gains directly support the effectiveness.
> ***

---

> > ### Comment · Reviewer_jRLg · 2025-11-26
> >
> > Thanks for the rebuttal, I will keep my score.

---

### Official Review · Reviewer_VFqG · 2025-11-01

**Soundness:** 3
**Presentation:** 4
**Contribution:** 3
**Rating:** 6
**Confidence:** 4

**Summary:**

This paper presents SR-3D, a vision-language model designed to unify spatial reasoning for both single-view 2D images and multi-view 3D data. The core mechanism involves enriching 2D visual features with 3D positional embeddings. These embeddings are derived from estimated (for 2D) or ground-truth (for 3D) depth and canonicalized into a shared 3D coordinate system, allowing a base 2D VLM to leverage its priors for complex 3D tasks. The model also introduces a "dynamic tiling-based region extractor" to efficiently process high-resolution region prompts. The authors demonstrate state-of-the-art performance on several 3D scene understanding and spatial reasoning benchmarks. They also introduce a new benchmark, SR-3D-Bench, to evaluate region-level spatial question answering in 3D environments.

**Strengths:**

1. The model's architecture effectively unifies single-view and multi-view spatial understanding. By mapping both to a canonical 3D positional space, the model successfully leverages powerful 2D pre-trained priors for 3D reasoning tasks.
2. The paper proposes a dynamic tiling-based region extractor. This seems to be an effective solution for handling high-resolution inputs and flexible region prompts (boxes, masks) without the distortion common in other methods, which is a practical and important problem.
3. The empirical evaluation is thorough and very strong. The authors achieve state-of-the-art results across multiple challenging benchmarks (ScanQA, SQA3D, VSI-Bench), outperforming numerous existing 2D and 3D models.
4. The introduction of the SR-3D-Bench benchmark is a good contribution. It fills a clear gap by providing a testbed for region-level spatial reasoning, which is more complex and ambiguous than global-level QA.

**Weaknesses:**

1. The model's capability is currently restricted to static scenes. The authors acknowledge this in the limitations, but it is a significant point, as many target applications for 3D spatial reasoning (like robotics or AR/VR) involve dynamic environments with moving objects. The proposed positional encoding scheme does not inherently account for temporal dynamics.
2. The ablation study in Table 8, while justifying the full model, also shows that the 3D positional embeddings ("3D PE") alone (without 2D pre-training) offer limited gains. The authors' conclusion that this is due to scale and that "larger-scale settings" would better exploit them is speculative. A deeper analysis of why the 3D PE isn't more impactful on its own would strengthen the paper.
3. The paper notes a slight but consistent performance degradation on OCR-related tasks. This suggests that the 3D-aware pre-training stage, while boosting spatial skills, may mildly compromise some of the base VLM's other established capabilities.

**Questions:**

1. The authors mention that a single, unified checkpoint for both 2D and 3D tasks is left for future work. What are the primary difficulties in co-training the model on both the 2D pre-training and 3D fine-tuning datasets? Does it introduce optimization challenges or data balancing issues?
2. What is the computational and memory overhead of the dynamic tiling-based region extractor?

---

> ### Author Response · Authors · 2025-11-24
> **Response to Reviewer VFqG (1/2)**
>
> We thank the reviewer for their valuable feedback. We address your comments as follows.
> ***
> >**W1:** *SR3D is restricted to static scenes and does not inherently account for temporal dynamics.*
>
> **A1:** While our main experiments focus on static 3D benchmarks, our design naturally extends to dynamic environments: each frame’s pointmap can be interpreted as a 4D pointmap, allowing the positional encoding to capture spatiotemporal structure without any architectural changes.
>
> To validate this, we conduct experiments on VLM4D, a benchmark explicitly designed to evaluate spatiotemporal reasoning in VLMs and which also provides a supervised training set for fine-tuning. The results below show that fine-tuning SR-3D on the VLM4D training set yields clear improvements, and adding positional encoding further boosts 4D reasoning performance.
>
> \begin{array}{l c}
> \hline
> \textbf{Method} & \textbf{Accuracy} \newline
> \hline
> \text{Qwen2.5-VL-7B} & 43.80 \newline
> \text{SR-3D-8B (trained on static-only, zero-shot)} & 31.76 \newline
> \text{NVILA-Lite-8B (finetuned, w/o PE)} & 51.76 \newline
> \text{SR-3D-8B (finetuned, w/ PE)} & 55.69 \newline
> \hline
> \end{array}
> ***
>
> >**W2:** *Why the 3D PE isn't more impactful on its own.*
>
> **A2:** 3D PE alone brings limited gains, and this pattern is consistent with prior work. LLAVA-3D reports only small improvements when injecting 3D positional cues into 2D patches (+0.4 on ScanQA and +0.3 on SQA3D), and Video3D-LLM even shows a slight drop (–0.41 on ScanQA when using learnable parameters). Both works only see noticeable gains when evaluating tasks that directly use coordinates as input (e.g., Scan2Cap). Since we treat Scan2Cap through image–region inputs instead of explicit coordinates, we do not observe such gains on Scan2Cap either. Together, these results suggest that 3D positional signals by themselves are not sufficient to learn strong 3D representations for general tasks.
>
> Our findings show that the key limitation is the scale of supervision. 3D PE includes learnable parameters in the MLP, but the available 3D annotations contain only about 6K unique scans, which is not enough for the model to learn meaningful geometric structure. To address this, we align the camera space so the model can benefit from million-scale single-view RGB relative-depth pretraining, which provides large-scale geometric supervision. When combined with 3D PE, this setup yields clear performance gains and demonstrates that **positional encoding becomes effective when supported by sufficient data and an aligned training design**.
> ***
>
> >**W3:** *Performance degradation on OCR-related tasks.*
>
> **A3:** Due to limited computational resources and the compressed training schedule, we were only able to include ~50% of the OCR-related instruction-tuning samples. Despite this reduction, the performance gaps on TextVQA, ChartQA, and DocVQA remain **under 1%**. We expect that incorporating the full SFT dataset would eliminate this small degradation.
> ***

---

> ### Author Response · Authors · 2025-11-24
> **Response to Reviewer VFqG (2/2)**
>
> >**Q1:** *Difficulties in co-training 2D and 3D datasets.*
>
> **A4:** Training a unified model on both 2D images and multi-view 3D videos is far more challenging than simply mixing the datasets. A full co-training cycle currently takes about 20 days on 16 nodes. This is mainly due to the much longer token sequences produced by multi-view videos, which reduce the batch size to roughly one quarter of the 2D-only setup and substantially increase training cost. Additionally, the strong imbalance between 2D and 3D data requires careful tuning of sampling ratios to prevent the smaller 3D set from being overshadowed, making a naïve “YOLO-style” run unlikely to succeed.
>
>
> Despite these constraints, we still ran a co-training experiment within our limited budget using subsets of the data. The results show that co-training is feasible and achieves competitive performance across both 2D spatial QA benchmarks (BLINK(S+D), SAT, EmbSpat, RWQA), general QA benchmarks (POPE, MMMU), and 3D video tasks (VSI-Bench). The co-trained model matches the specialized SR-3D variants on 2D tasks and even improves the 3D video performance (64.8 vs. 62.9), suggesting that the main barrier to a unified checkpoint is computational cost rather than conceptual difficulty.
>
> \begin{array}{lcccccc}
> \hline
> \bf{Model} & \bf{BLINK(S{+}D)} & \bf{SAT} & \bf{EmbSpat} & \bf{POPE} & \bf{MMMU} & \bf{VSIBench} \newline
> \hline
> \text{NVILA-Lite-8B} & 76.5 & 62.6 & 68.9 & 88.1 & 25.1 & - \newline
> \text{SR-3D (2D / 3D)} & 81.0 & 64.0 & 72.5 & 87.6 & 24.6 & 62.9 \newline
> \text{SR-3D (Co-train)} & 80.6 & 69.3 & 71.5 & 87.3 & 25.7 & 64.8 \newline
> \hline
> \end{array}
> ***
>
> >**Q2:** *Computational and memory overhead of the dynamic tiling-based region extractor.*
>
> **A5:** The region extractor introduces only lightweight intermediate tensors of size $O(M \cdot \tilde{L})$, where $M$ denotes the number of region masks and $\tilde{L}$ is the post-tiling token count. Consequently, its memory and compute overhead are negligible relative to the backbone; nearly all additional cost stems from the increased number of visual tokens introduced by tiling. Implementation-wise, the tile-then-stitch extractor is efficient: it operates directly on the token grid of length $L$ (rather than full-resolution pixels), batches all region masks per sample into a single, and performs pooling using a single GPU-efficient *einsum* over concatenated features, thereby avoiding per-region encoder passes. Empirically, the region extraction step consistently takes less than 1 ms on a single A100 GPU and contributes only a small fraction of the end-to-end inference time.
>
> \begin{array}{lcc}
> \hline
> \bf{Setting} & \bf{Total\ Inference\ Time\ (s)}& \bf{Region\ Extract\ Time\ (s)} \newline
> \hline
> \text{w/o tiling} & 0.221 & 0.000375 \newline
> \text{w/ tiling (512 x 512 → 2 x 2 +1 tiles)} & 0.269 & 0.000529 \newline
> \text{w/ tiling (1024 x 1024 → 3 x 3 +1 tiles)} & 0.459 & 0.000757 \newline
> \hline
> \end{array}
> ***

---

> > ### Comment · Reviewer_VFqG · 2025-11-26
> >
> > Thank you for the detailed response, which addresses most of my concerns. I will keep my original rating.

---

### Meta-Review · Area_Chair_kEDq · 2025-12-26

**Summary:**

Reviewers confirmed the novelty, thorough experiments and writing of this paper. After rebuttal and discussion, most of the concerns raised by reviewers are resolved the authors. I thereby recommend to accept this submission.

**Reviewer Concerns:**

The major concerns include optimization challenges, computational overhead, more evidences to support unified representation, and ablation studies. During the rebuttal, the authors explained how to solve the optimization challenges, and provided more evidence to support unified representation. They also clarified the evidences (e.g. Table 8) to support the unified representation.

**Reviewer Scores:**

According to the rebuttal history, the score 4 raised to a 6 (and reversed to 4).

---

### Decision · Program_Chairs · 2026-01-26

Accept (Poster)